

**Mercury fluxes, budgets and pools in forest ecosystems of China: A critical review**
Jun Zhou [a, c, d, ]*, Buyun Du [b, e], Zhangwei Wang [d, e], Lihai Shang [b], Jing Zhou [a, c, ]*
a. Key Laboratory of Soil Environment and Pollution Remediation, Institute of Soil Science, Chinese Academy
of Sciences, Nanjing 210008, China.
b. State Key Laboratory of Environmental Geochemistry, Institute of Geochemistry, Chinese Academy of
Sciences, Guiyang 550081, China.
c. National Engineering and Technology Research Center for Red Soil Improvement, Red Soil Ecological
Experiment Station, Chinese Academy of Sciences, Yingtan 335211, China.
d. Research Center for Eco-Environmental Sciences, Chinese Academy of Sciences, Beijing 100085, China.
e. University of Chinese Academy of Sciences, Beijing 100049, China
Correspondence to: Jun Zhou (zhoujun@issas.ac.cn); Jing Zhou (zhoujing@issas.ac.cn)



**Abstract:** Mercury (Hg) accumulation and retention in forest ecosystems play a key role in global biogeochemical
cycling of Hg. Especially in China, forests are suffering highly elevated Hg loads. Numerous studies have been
conducted to characterize the fluxes and pools of Hg in the terrestrial forests in China during the past decade, which
provide insights into spatial distributions and estimate the Hg mass balance in forests through observations at widely
diverse subtropical and temperate locations. In this paper, we present a comprehensive review of the research status
of forest Hg in China to characterize the Hg budgets and pools. Averaged total Hg (THg) inputs at remote forests and
rural & suburban forests in China are about 2 to 4-fold and 2.5 to 5-fold higher than the observed values in Europe
and North America, respectively. The highly elevated THg inputs are mainly derived from the elevated atmospheric
Hg concentrations. Additionally, production of litterfall biomass is showed to be an important influential factor
raising the high Hg inputs at subtropical forests. Compared to the input, THg outputs from the forest ecosystems are
relative small, which results in large amount of Hg resided in the forest soils. The annual THg retentions range from
26.1 to 60.4 μg m$^{-2}$ at subtropical forests and from 12.4 to 26.2 μg m$^{-2}$ at temperate forests of China, which are about
3.8- to 7.9-fold and 1.2 to 2.8-fold higher compared to those in North America. Given the large areal coverage, THg
retention in forest is appropriately 69 t yr$^{-1}$ in China and is much high than that in global scale estimated by models.
The much higher THg retention has elevated the THg pools in Chinese subtropical forests, which poses a serious
threat for large Hg pulses remitted back to the atmosphere and additional ecological risks in the forest. The current
study has implication for the role of China forests in the global Hg biogeochemical cycle and the optimization of
atmospheric Hg transport and deposition models.
**Keywords:** Trace metals; Atmospheric deposition; Input–output; Storage; Risk assessment



## 1. Introduction

Mercury (Hg) is considered as a highly toxic heavy metal due to its biogeochemical properties and its toxicity that can affect the health of human and ecosystems (Kojta et al., 2015; Falandysz et al., 2015a; Zhou et al., 2015b; Du et al., 2016). Unlike other heavy metals, atmospheric Hg may exist three operational Hg forms: gaseous elemental Hg (GEM); gaseous oxidized Hg (GOM, also known as reactive gaseous Hg); and particulate-bound Hg (PBM) (Fu et al., 2015). Due to its strong stability and low water solubility, GEM is the most abundant (more than 90%) in the atmosphere and has long resistance time of 0.5–2 years, which can be transported globally and deposited to the remote eco-environment (Gustin et al., 2012; Åkerblom et al., 2015). Atmospheric Hg deposition to terrestrial and aquatic ecosystems plays a significant role in the global biogeochemical cycling of Hg (Zhou et al., 2103a; Blackwell and Driscoll, 2015). Consequently, an understanding of how Hg is transported, deposited and circled the globe is significant for a full understanding and quantifying of Hg biogeochemical cycles (Fisher and Wolfe, 2011).

Atmospheric Hg is nearly the exclusive source of Hg in forest biomass due to the limitation of root uptake (Grigal, 2003). The forest canopy is a major receptor of Hg in terrestrial forest ecosystems, which can absorb Hg through stomatal uptake of GEM, and adsorb PBM and GOM onto foliage surface (Fu et al., 2015). Therefore, atmospheric deposition by litterfall and throughfall is the largest input of Hg to forested watersheds that are not affected by natural geologic or point sources (Blackwell and Driscoll, 2015; Zhou et al., 2016b). Forest ecosystems are considered as a large sink of atmospheric Hg and an active pool of Hg, which is a vital part in the global Hg cycle (Friedli et al., 2007; Zhou et al., 2016a; Ma et al., 2016). Additionally, the large amounts of Hg inputted to the forest are sequestrated in the vegetation and soils, and have produced ecological risks on the bioaccumulation of Hg in the internal forest. For example, Hg sequestrated in the forest soil are considered as potential sources of both total Hg (THg) and methylmercury (MeHg) to downstream aquatic ecosystems (Selvendiran et al., 2008; Ma et al., 2015). Moreover, Hg in the forest soil and biomass can be directly used by forest animals that may be highly vulnerable to the increasing Hg loads (Rimmer et al., 2010; Zhou et al., 2016a).

Dynamic and budget studies quantifying Hg flux and pool in the terrestrial forests are necessary for investigating status of Hg inputs to, retention within, and output from forest ecosystems. Many studies have improved our knowledge of current Hg pools and fluxes (Larssen et al., 2008; Ma et al., 2015; Grigal et al., 2000; Grigal, 2003). However, terrestrial forest has constantly been underestimated as sink for atmospheric Hg on a global scale (Wang et al., 2016b; Obrist, 2007). Previous reviews were mainly focused on the atmospheric Hg concentrations (Fu et al., 2015), atmospheric Hg depositions (Wang et al., 2016a; Wright et al., 2016) and air–surface fluxes (Zhu et al., 2016);





however, no studies aimed at the Hg budgets and quantified the Hg retention in the forest ecosystems. Agnan et al.
suggested that the earth's surface contributed to half of the global natural emissions (607 Mg yr$^{-1}$); however, the
estimated value had a large uncertainty ranges between $-513$ to 1353 Mg yr$^{-1}$, due to what degree forests are net
sinks or sources of GEM. China, the largest emitting country of anthropogenic Hg source, has done quite a lot work
to positioning the role of forests in the regional- and global-scale Hg biogeochemical cycles. In order to provide a
better understanding of current knowledge with respect to forest Hg in China and quantify the forest act as net sinks
or sources of GEM, we comprehensively review the forest Hg data in China to estimate the Hg mass balance in
forests based on the observations. The important ecological risk of Hg accumulation and storage in forest is also
presented. The Hg budgets in forests partly help dissolve the question: what degree the ecosystems are net sinks or
sources of atmospheric Hg. The implications and future research needs for further understanding of forest Hg in
China are also presented.

**2. Processes of Hg input**
2.1. Wet input

The THg and MeHg input fluxes by precipitation, throughfall and litterfall in forested area of China are showed

in Table 1. The averaged THg and MeHg concentrations in precipitation sampled via wet-only precipitation sampling
device at remote forests were 4.5 ng L$^{-1}$ ($n = 4$, range from 3.0 to 7.4 ng L$^{-1}$) and 0.06 ng L$^{-1}$ ($n = 2$, range from 0.04
to 0.08 ng L$^{-1}$) , respectively. Prospectively, the mean THg and MeHg concentrations in bulk precipitation samples
at remote forests of China were 12.5 ng L$^{-1}$ ($n = 3$, range from 9.9 to 14.2 ng L$^{-1}$) and 0.16 ng L$^{-1}$ ($n = 1$), which
were much higher than those collected by wet-only precipitation sampling devices (Table 1). Although the PBM and
GOM in remote forests were relatively lower, dry deposition of PBM and GOM can also contribute to the elevation
of Hg concentrations in bulk precipitation. At rural & suburban forests, the THg and MeHg concentrations were much
higher in wet-only precipitation, with the average concentration range from 10.9 to 32.3 ng L$^{-1}$ ($n = 5$, mean = 18.1
ng L$^{-1}$) and range from 0.20 to 0.24 ng L$^{-1}$ ($n = 2$, mean = 0.22 ng L$^{-1}$), respectively. Wet-only input fluxes of THg
and MeHg were comparable and ranged from 5.4 to 6.1 µg m$^{-2}$ yr$^{-1}$ ($n = 4$, mean = 5.8 µg m$^{-2}$ yr$^{-1}$) and 0.06 to 0.14
µg m$^{-2}$ yr$^{-1}$ ($n = 2$, mean = 0.10 µg m$^{-2}$ yr$^{-1}$) at remote sites, and ranged from 14.4 to 29 µg m$^{-2}$ yr$^{-1}$ ($n = 5$, mean =
18.1 µg m$^{-2}$ yr$^{-1}$) and 0.26 to 0.36 µg m$^{-2}$ yr$^{-1}$ ($n = 2$, mean = 0.31 µg m$^{-2}$ yr$^{-1}$) at rural & suburban forests,
respectively (Table 1). THg concentrations in precipitation and corresponding wet deposition fluxes at rural &
suburban forested areas were elevated compared to those in North America and Europe, but the concentrations and



93 fluxes at remote forests were in the lower range of those obtained from remote forested areas in North America and

94 Europe (Choi et al., 2008; Graydon et al., 2008; Åkerblom et al., 2015; Guentzel et al., 2001).

95  Previous studies suggested that THg in rainwaters was originated from the scavenging of PBM and GOM in the

96 atmosphere (Guentzel et al., 2001, Zhou et al., 2013a). Additionally, Fu et al. (2015) reviewed the THg fluxes in

97 China and observed significant correlations between rainwater THg concentrations and GOM as well as PBM

98 concentrations at urban, suburban and remote areas. However, THg concentrations in precipitations were not

99 significantly correlated with the three Hg forms of GEM, PBM and GOM in the forested areas of China (n= 10, 4

100 and 4; T test, p> 0.05 for all). The reason may be that reduced PBM and GOM in forested areas resulted in low

101 scavenging during wet deposition events (Lee et al., 2001; Seigneur et al., 2004). One the other hand, the vast majority

102 of forest were at high altitude with low-level clouds, which limited the scavenging height and reduced the washout

103 efficiency.

105 2.2. Throughfall and litterfall input

106  Throughfall and litterfall depositions are the two major pathways for Hg delivery to forest floor. Throughfall is

107 rainfall that delivers to the forest floor after interacting with the forest canopy, which can wash off a large portion of

108 the PBM and RGM deposited to forest leaves (Rea et al., 2000), resulting in higher THg and MeHg concentrations

109 compared to those in precipitation. There are many factors influencing THg concentrations and depositions by

110 throughfall, including canopy type (Demers et al., 2007; Åkerblom et al., 2015), meteorological conditions

111 (Blackwell and Driscoll, 2015b) and sample locations (Luo et al., 2015a). In addition, THg concentrations in

112 precipitations also significantly affected these in throughfall duo to similar source in both aqueous, which showed

113 significant positive correlations ($n = 9$, $r^2=0.92$, p<0.01). The THg concentrations were ranged from 8.9 to 40.2 ng

114 $L^{-1}$ ($n = 3$, mean = 28.6 ng $L^{-1}$) at remote forests and ranged from 20.1 to 69.7 ng $L^{-1}$ ($n = 6$, mean = 42.5 ng $L^{-1}$) at

115 rural & suburban forests, which averaged 2.6- and 2.0-fold compared to the corresponding THg concentrations in

116 precipitation (Table 1).

117  The mean THg depositions by throughfall were 36.3 µg $m^{-2}$ $yr^{-1}$ (rang of 10.5−57.1 µg $m^{-2}$ $yr^{-1}$) at remote

118 forests and 42.5 µg $m^{-2}$ $yr^{-1}$ (rang of 21.8−71.3 µg $m^{-2}$ $yr^{-1}$) at rural & suburban forests, respectively. The means of

119 THg inputs are 2–3 times and 4–6 times higher than those of the European values (mean = 19.0 µg $m^{-2}$ $yr^{-1}$) and the

120 North America values (mean = 9.3 µg $m^{-2}$ $yr^{-1}$), the ranges of which were between 12.0 and 40.1 µg $m^{-2}$ $yr^{-1}$ and

121 between 2.07 and 25.4 µg $m^{-2}$ $yr^{-1}$, respectively (Fig. 4). At forests of China, throughfall contributed higher Hg inputs



than those of wet inputs, with throughfall ranging from about 1.7 to 2.5 times the wet input (Fig. 1). However, these
were different with the North America forests, where throughfall Hg inputs were found to be lower than wet-only
depositions in deciduous forests, but to be higher than wet-only depositions in coniferous forests (Wright et al., 2016).
Litterfall Hg inputs have been confirmed to be the other important pathway trapping atmospheric Hg to the
forest floor via senesced leaves, needles, twigs, and branches, and other plant tissues. Concentrations of Hg in litterfall
could be affected by many factors, such as tree species, lifespan, and environmental factors (e.g., solar irradiation,
air temperature, altitude, etc.) (Ericksen and Gustin, 2003; Poissant et al., 2008; Blackwell and Driscoll, 2015b; Zhou
et al., 2017a). However, atmospheric Hg concentrations play the most important role in Hg concentrations in litterfall,
and Hg concentrations in atmosphere were deemed to be a good indicators of leaf Hg contents in forest areas (Fay
and Gustin, 2007; Niu et al., 2011). Based on the available atmospheric total gaseous Hg (TGM) or GEM
concentrations and litterfall Hg concentrations in 11 forested areas and 14 pairs of datasets in China, annual mean
atmospheric TGM/GEM concentrations were significantly correlated with the THg concentrations in litterfall
samples (Fig. 2). The significant correlation might verify that foliage can effectively trap Hg from the atmosphere by
accumulation Hg through stomatal uptake of GEM (Fay and Gustin, 2007; Fu et al., 2010a, b; Laacouri et al., 2013;
Zhou et al., 2017b). The mean THg and MeHg concentrations in litterfall at remote sites ranged from 12.6 to 135.1
ng g$^{-1}$ (mean = 54 ng g$^{-1}$, n = 12) and from 0.28 to 0.48 ng g$^{-1}$ (mean = 0.38 ng g$^{-1}$, n = 2), respectively (Table 1).
Such litterfall THg and MeHg concentrations were higher in rural & suburban areas, with mean concentration range
of 25.8 to 176.1 ng g$^{-1}$ (mean = 61.2 ng g$^{-1}$, n = 5) and 0.21 to 0.84 ng g$^{-1}$ (mean = 0.52 ng g$^{-1}$, n = 4), respectively.
THg and MeHg concentrations in litterfall at rural & suburban areas of China were higher than those in North America
and Europe, but litterfall concentrations of THg and MeHg at remote areas were compared those observed in North
America and Europe, except in Mt. Leigong, Guizhou Provence (Table 1, Fig. S1). Although Mt. Leigong was
relatively isolated from anthropogenic activities with lower GOM, PBM, precipitation and throughfall Hg
concentrations, GEM could undergo long-range transport from emission sources. The GEM concentration was 2.80
ng m$^{-3}$ in Mt. Leigong that is about 170 km to the large Hg mine of Wanshan (Fu et al., 2010a). The relatively higher
GEM concentration resulted in elevated litterfall Hg concentrations.
Mean THg inputs by litterfall from 20 forests in China (41.8 µg m$^{-2}$ yr$^{-1}$) were approximately 2 to 3 times higher
than those in Europe over 11 sites (14.2 µg m$^{-2}$ yr$^{-1}$) and more than 3 times higher than those in North America over
37 sites (12.9 µg m$^{-2}$ yr$^{-1}$) (Fig. 4). Since litterfall THg inputs to terrestrial ecosystems are estimated by multiplying
the biomass and corresponding THg content in litterfall, both of them could influence the input fluxes. Therefore,



compared to North America and Europe, higher TGM or GEM concentrations in rural & suburban forests of China
resulted in the elevated litterfall Hg concentrations and corresponding higher fluxes in China. However, it should be
noted that the litterfall biomass productions in forests of China ($565 \pm 450$ g m$^{-2}$ yr$^{-1}$) were more than 2-fold higher
than those observed in North America and Europe ($200 \pm 145$ g m$^{-2}$ yr$^{-1}$). The regional differences of litterfall Hg
inputs to forest ecosystems was primarily resulted by the factor of litterfall biomasses rather than litterfall Hg
concentrations, as evidenced by the much stronger correlation between litterfall Hg input fluxes and litter biomass
productions than that with litterfall THg concentrations (Fig. 3a and b).
The total Hg input as the sum of Hg input by litterfall and throughfall (i.e., input flux by litterfall + input flux
by throughfall) to forests were ranged from 47.7 to 291.3 $\mu$g m$^{-2}$ yr$^{-1}$ (n=11 from 9 forests) in China (Fig. 1). Here,
it should be noted that the highest Hg deposition (291.3 $\mu$g m$^{-2}$ yr$^{-1}$) was observed at Tieshanping forest from March
2005 to March 2006 (Wang et al., 2009); however, due to overestimation of litterfall biomass, the measured Hg fluxes
were more than 3 times the recent studies by Luo et al. (2015a) in 2010–2011 and Zhou et al. (2017b) in 2014 –2015.
The much higher Hg input at Tieshanping forest is due to it located near the center of Chongqing City (20 km), the
annual atmospheric emissions of which just from coal combustion was 4.97 t (Wang et al., 2006) and Hg pollution
was regarded as major environmental burdens in Chongqing (Yang et al., 2009). If we use the updated Hg inputs
fluxes by Luo et al. (2015b) at Tieshanping forest, the annually mean total Hg input flux was 73.9 $\mu$g m$^{-2}$ yr$^{-1}$ (n=10)
in China. Hg input to forest floor via litterfall was substantially comparable or greater than the throughfall input and
the litterfall to throughfall input ratios range from 0.33 to 6.59 (mean= 2.14), indicating that Hg input via litterfall
surpassed that by throughfall and become the major pathway of Hg input to forests in China. The observed ratios in
forest ecosystem of China were much greater than those observed in North America and Europe. Ratios of litterfall
Hg input to throughfall Hg input to forest ecosystems were in the range of 0.27 to 1.56 (mean=0.89; n = 9) in Europe
(Schwesig and Matzner, 2000; Hultberg et al., 1995; Iverfeldt et al.,1991; Larssen et al., 2008; Lee et al., 2000;
Munthe et al., 1995, 1998; Schwesig and Matzner, 2001), and in the range of 0.60 to 4.13 in North America (mean =
1.37; n = 16) (Blackwell and Driscoll, 2015b; Choi et al., 2008; Demers et al., 2007; Kalicin et al., 2008; Kolka 1999;
Grigal et al., 2000; Lindberg et al., 1994; Fisher and Wolfe, 2011; Rea et al., 1996, 2001; Johnson, 2002; Johnson et
al., 2007; Nelson et al., 2007; St. Louis et al., 2001; Graydon et al., 2008), which was about 2.4 to 1.6 times lower
compared to the ratios observed in China. The reason is the much higher litterfall biomass production in forest of
China as we stated above.
Additionally, more than 90% of Hg in litterfall biomass is considered to be uptake from atmosphere, and





throughfall can wash off most of the PHg and RGM on the leaf surface by previous dry depositions; therefore, litterfall
and throughfall Hg inputs could be a good indicator of TGM dry deposition to forest ecosystems (Gustin et al., 2012;
Zhou et al., 2013a; Fu et al., 2015). Considering dry Hg input in a forest ecosystem as the difference between total
Hg input and wet Hg input (dry Hg input = total Hg input – wet Hg input), more than 80% of total Hg inputs were
from dry inputs in forests of China, which was higher than those in North America and Europe (70%) but lower than
those in Brazil (85%) (Wang et al., 2016).

Higher dry and wet depositions resulted in higher total Hg inputs to Chinese forests, which averaged 78.4 µg

$m^{-2}$ $yr^{-1}$ at remote forests and 106.5 µg $m^{-2}$ $yr^{-1}$ at rural & suburban forests, and ranged from 47.7 to 119.5 µg $m^{-2}$
$yr^{-1}$ (n= 5) and from 56.0 to 291.3 µg $m^{-2}$ $yr^{-1}$ (n= 6), respectively. We have also reviewed the THg inputs by
throughfall and litterfall in the Europe and North America (Fig. 4), and the results showed that THg inputs were
significantly lower than those observed in China (p<0.05 for Europe and p<0.01 for North America). Mean THg input
was about 39.2 µg $m^{-2}$ $yr^{-1}$ (n= 9) in the Europe, which was about 2.0- and 2.5-fold lower than that observed at
remote forests and rural & suburban forests in China. Even lower THg input was found in the North America (20.2
µg $m^{-2}$ $yr^{-1}$, n= 17) and was about 4- and 5-fold lower than that at remote forests and rural & suburban forests in
China.

**3. Processes of Hg output**
3.1. Exports from surface runoff and underground runoff

The dominate pathways of Hg output from forest catchments were runoffs and soil-atmosphere exchange fluxes.

The output fluxes of THg and MeHg via surface runoff measured in China are showed in Table 2. The mean THg and
MeHg concentrations in surface runoff ranged from 2.3 to 17.2 ng $L^{-1}$ (mean = 6.0 ± 4.1 ng $L^{-1}$, n = 11) and from
0.2 to 0.25 ng $L^{-1}$ (mean = 0.23 ng $L^{-1}$, n = 2), respectively. Comparing to the THg (40.5± 19.6 ng $L^{-1}$) and MeHg
(0.32 ng $L^{-1}$) in throughfall, the corresponding Hg concentrations in surface runoffs were seemed much lower, which
was consistent with the general concept that forests had the filtering function between atmosphere and hydrosphere
(Ericksen et al., 2003; Larssen et al., 2008). The export fluxes of THg via surface runoffs and/or stream waters ranged
from 3.0 to 8.6 µg $m^{-2}$ $yr^{-1}$ (mean = 4.8 ± 2.6 µg $m^{-2}$ $yr^{-1}$, n = 6). Luo et al. (2014) collected 117 stream water samples
in China, including 42 streams from 9 sites in the northeastern forests and 75 streams from 16 sites in the southern
forests, and the result showed that THg concentration was higher in northeastern forests (17.2 ± 11.0 ng $L^{-1}$) than
that in the southern forests (6.2 ± 6.4 ng $L^{-1}$). The THg concentrations in stream water were positively correlated to





209 DOC concentrations, suggesting that the DOC may facilitate the Hg mobility. Due to cool and dry climate in northern

210 forests, litter decomposed more slowly and resulted in deeper litter and organic layers than those in southern forests

211 (Zhou et al., 2015a, 2017a). Therefore, soil erosion in northern forests with higher DOC in stream waters resulted in

212 higher THg concentrations.

213  No statistically significant correlations were showed between THg concentrations in stream water and

214 throughfall ($r^2 = 0.00$, p>0.05, n = 9), and between throughfall Hg inputs and stream water exports ($r^2 = 0.03$, p>0.05,

215 n = 6), implying that THg output from stream water was regulated directly by processes other than current deposition

216 input in these forested catchments. However, THg export fluxes via runoff and/or stream waters were significantly

217 correlated with THg concentrations in surface soils (organic layer or top 10 cm) ($r^2 = 0.52$, p < 0.05, Fig. S2). Higher

218 THg depositions have resulted in much higher soil THg concentrations at forest sites of China. Although soils in

219 forests have been suggested as filters between throughfall and stream waters, but THg in stream waters also can

220 desorb from soils (Xue et al., 2013). Yin et al. (1997) suggested that higher Hg concentrations in the water of

221 prefiltration and soils both could be resulted in higher Hg concentrations in the leachate. Therefore, higher soil Hg

222 contents caused by higher deposition at forests of China caused high Hg concentrations in the stream water. Since

223 the adsorption and desorption of THg in soils cloud also depend on other factors, including the soil physical and

224 chemical properties (pH, organic matter, consistency) and leachate properties (pH, dissolved organic matter, salinity)

225 (Yin et al., 1997; Xue et al., 2013; Liao et al., 2009), the deduction may have large uncertainties.

226  The direct measurements of THg in underground runoffs were not conducted in any forests of China, but they

227 played important roles in the THg export from forests due to both of the amounts and THg concentrations usually

228 higher than those of surface runoffs in subtropical forests (Liu, 2005; Luo et al., 2015b). Several studies have

229 measured THg concentrations in solutions of soil profiles in subtropical forest of Tieshanping, which was averaged

230 21.8 ng L$^{-1}$ and ranged from 1.98 to 60 ng L$^{-1}$ (Wang et al., 2009; Zhou et al., 2015; Luo et al., 2015b). The observed

231 THg concentrations of soil solution was higher than those in five Swiss forest soils, and the reason may be due to

232 higher THg loads and soil THg content in this Chinses forest. Although no studies directly measured the export flux

233 of THg via underground runoff, we roughly estimated the flux based on the THg in soil solutions and runoff amount

234 in Tieshanping forest, which is 6.0 μg m$^{-2}$ yr$^{-1}$; therefore, the total Hg output by runoffs as the sum of Hg output by

235 surface runoff (3.5 μg m$^{-2}$ yr$^{-1}$) and underground runoff (6.0 μg m$^{-2}$ yr$^{-1}$) was 9.5 μg m$^{-2}$ yr$^{-1}$.


237 3.2. Export of soil-atmosphere exchange fluxes



Table 3 shows the statistical summary of soil-atmosphere Hg exchange fluxes and associated site information
in the 30 forest sites. Mean soil-atmosphere Hg exchange fluxes at remote forests were in the range of 1.6–4.77 ng
$m^{-2}$ $hr^{-1}$ (mean = 3.3 ± 3.4 ng $m^{-2}$ $hr^{-1}$, n = 12), and those at rural & suburban forests were significantly higher (T
test, $p < 0.05$) and ranged from –0.8 to 17.8 ng $m^{-2}$ $hr^{-1}$ (mean = 8.3 ± 7.1 ng $m^{-2}$ $hr^{-1}$, n = 18). Generally, soil-
atmosphere Hg exchange fluxes are bi-directional. Nevertheless, only one site showed overall net deposition of −0.8
ng $m^{-2}$ $hr^{-1}$ in the wetland of Tieshanping forest and the other forest soils showed overall net emissions in China.
Many studies have identified factors that correlate with the magnitude and direction of soil-atmosphere Hg
exchange fluxes, including atmospheric and soil physicochemical properties. The well-known factors studied in the
previous researches influencing soil-atmosphere Hg exchange fluxes included substrate Hg concentration, air and
soil temperature, measurement methodology, as well as environmental variables (e.g. forest type, terrain type and
soil cover). The most commonly promoting $Hg^0$ production is solar radiation that is reported with positive correlations
in all the studied forests in China (n = 30). The relationship is mainly attributed to photochemical reduction of soil-
bound Hg, which converts soil $Hg^{2+}$ to volatile $Hg^0$ (Amyot et al., 1994, 1997; Carpi and Lindberg, 1997; Moore and
Carpi, 2005; Xin et al., 2007; Zhou et al., 2017b). Photo-reduction is a major driver of $Hg^0$ generation and evasion
from soils (Choi and Holsen, 2009; Engle et al., 2001; Zhou et al., 2015a, 2017b), although other abiotic and biotic
processes also resulted in translation of $Hg^{2+}$ to $Hg^0$ production, including reduction by humic acids (Alberts et al.,
1974; Allard and Arsenie, 1991) and iron oxides under anoxic conditions (Lin and Pehkonen, 1997) as well as
reduction by microorganisms (Siciliano et al., 2002; Agnan et al., 2016) and/or microbial exudates (Poulain et al.,
2007, 2004; Fritsche et al., 2008). Additionally, other important correlation was identified with soil or air temperature,
which is also significantly correlated to the $Hg^0$ production and observed with soil-atmosphere Hg flux in all the
forests in China (n=30). Soil temperature was generally stimulated directly to activation energy of $Hg^0$ (Gustin et al.,
1997; Edwards and Howard, 2013) or stimulation $Hg^0$ evasion by action of soil microorganism activity (Pannu et al.,

2014).

Agnan et al. (2016) showed that substrate Hg concentration was significantly correlated with soil-atmosphere
Hg fluxes across Hg-enriched sites by large global data set (n = 538), but an apparent lack of correlation between
substrate Hg concentrations and soil-atmosphere Hg fluxes across all background soils (n = 307) that defined as
substrate Hg concentrations ≤ 300 ng $g^{-1}$ and atmospheric $Hg^0$ concentrations ≤ 3 ng $m^{-3}$. Across all vegetation-
covered soils (forest and wetland) of China, the correlation between soil Hg concentrations and soil-atmosphere
exchange fluxes also did not show significantly across the entire database ($r^2 = 0.02$, $p > 0.05$, n = 25), which was





consistent with the global database set in background soils (Agnan et al., 2016). The lack of correlation between
substrate Hg concentrations and soil-atmosphere Hg fluxes may indicate either little control of soil Hg content on the
exchange fluxes across forested areas, or that other parameters prevailed over the effects of soil Hg content.
Alternatively, forest areas showed a much narrower range of soil Hg content compared to Hg-enriched substrates,
which influenced the fluxes inconspicuously. However, Zhou et al. (2016c) reported strongly positive correlations
between soil Hg contents and fluxes at individual forest of Tieshanping subtropical forest ($r^2$=0.97, p<0.001) due to
the sampling locations that were nearby and have similar other environmental factors.
According to the two-resistance exchange interface model, the exchange fluxes are caused by the gradient of
$Hg^0$ concentrations on both interfaces (Zhang et al., 2002); therefore, high $Hg^0$ concentrations in the atmosphere will
reduce the potential of $Hg^0$ produced in the soil and diffusion to atmosphere. Laboratory and filed simulation studies
showed that elevated atmospheric Hg concentrations significantly inhibited soil Hg volatilizations (Zhou et al., 2017b;
Ericksen and Gustin, 2004; Hanson et al., 1995; Poulain et al., 2004). Atmospheric compensation point for $Hg^0$ flux
was firstly presented by Hanson et al. (1995), which is the atmospheric Hg concentration at which no net flux occurs
between soil and air (flux to be 0). A previous study using the global database set in background areas showed
significant correlation between atmospheric Hg and soil-atmosphere exchange fluxes (p < 0.001, n = 263) (Agnan et
al., 2016). In contrast, based on the database combining all forest-covered soils in China, correlation between
atmospheric Hg concentrations and soil-atmosphere exchange fluxes was not significant ($r^2$ = 0.05, p > 0.05, n = 28),
which was inconsistent to the concept of the compensation point. The no correlation was contributed to the variations
of environmental factors and Hg emissions at forest sites that resulted in a different buildup of GEM/TGM near the
surface in the boundary layer. Thus, high soil emissions caused high GEM/TGM concentrations and not vice versa
via a control of air GEM/TGM concentrations on soil-atmosphere exchange fluxes. However, in individual forests,
studies showed that compensation points at subtropical forests were in the range of 3.89–6.90 ng m$^{-3}$ in Tieshanping
forest stands (Du et al., 2014; Zhou et al., 2016c) and 7.75 ng m$^{-3}$ in Qianyanzhou forest (Luo et al., 2015a), which
were much higher than that calculated according to the global database in background sites (2.75 ng m$^{-3}$, Agnan et
al., 2016). Higher compensation points observed in China also imply that natural surface contribute larger
atmospheric Hg pools in China.
Additionally, studies have observed predictable influences of environmental variables on $Hg^0$ exchange across
multiple forests when using consistent measurement methodology, such as significant correlations with air relative
humidity (Ma et al., 2013, 2015; Du et al., 2014; Luo et al., 2015a). However, it should be noted that the correlation



between air humidity and air temperature were also observed, indicating that air temperature may control the air and
soil humility. Furthermore, soil moisture stimulated soil Hg emissions at Qianyanzhou and Zhuzhou forests (Luo et
al., 2015a; Du et al., 2014) but reduced emissions at Tieshanping forest stands (Du et al., 2014; Zhou et al., 2016c).
Previous studies suggested that soil moisture contributed to TGM flux had optimum interval and should be under
intermediate conditions, neither under fairly dry nor very wet (Gustin and Stamenkovic, 2005; Lin et al., 2010; Pannu
et al., 2014; Obrist et al., 2014; Zhou et al., 2017b), which can elucidate the different correlations at different forest
ecosystems.

Fig. 5 shows the seasonal variations of soil-atmosphere Hg exchange fluxes at forest areas in China. The

variations can be classified into two distinct types: evergreen forest and deciduous forest. At evergreen forests, the
mean exchange fluxes in warm seasons (summer and spring) were relative higher than those in cold seasons (winter
and autumn, t test: $p < 0.05$ for all). Solar radiation over the forest canopy was much higher in the warm seasons, but
the branches and leaves were also luxuriant, so soils received similar sunlight with other seasons at the subtropical
evergreen forests (Ma et al., 2013). Therefore, elevated soil-atmosphere Hg exchange fluxes in warm seasons under
the evergreen canopy were mainly caused by the increasing soil/air temperature. In contrast, in the deciduous forests,
such as larch, mixed broadleaf forest and wetland in Mt. Dongling, the means of soil-atmosphere Hg exchange fluxes
were significantly higher in cold seasons (leaf-off period) than that in the other seasons (t test: $p < 0.01$). Solar
radiation was the maximum amount reaching the forest floor during leaf-off periods in winter, which was
approximately 300 W m$^{-2}$ and promoted Hg$^0$ production. Whereas during leaf-on periods in summer, the maximum
solar radiation at the forest floor was only about 116 W m$^{-2}$.

In summary, our results suggested that soil-atmosphere Hg exchange fluxes are highly dependent on temperature

at the evergreen forests, which increased the rate of reduction of Hg$^{2+}$ by thermal processes, biological activities and
stimulating Hg$^0$ evasion (Choi and Holsen, 2009; Engle et al., 2001; Poissant et al., 1998; Zhang et al., 2001). In the
deciduous forests, the fluxes were similar to evergreen forests during leaf-on periods, whereas the exchange fluxes
are dependent on solar radiation during leaf-off periods because that can directly reach to the forest floor. Although
soil received direct solar radiation at forests in north China during leaf-off periods that can be lasted for about half a
year (November to April), the exchange fluxes displayed a spatial pattern with significantly lower fluxes in the
temperate zones in north China than those at subtropical zones in south China (t test, $p<0.01$) due to lower temperature
at temperate zones. Additionally, the remote forests in the temperate zones in north China had similar exchange fluxes
to Europe and North America, due to similar forest type, soil properties, TGM concentrations and environmental





factors at those forests. However, the fluxes at subtropical zones of remote, rural & suburban forests were generally
higher compared to those observed in North America, Europe and South America. The reason may be that forest soils
at these areas have higher THg concentrations and receive more solar radiation and causing higher temperature than
those at boreal and temperate forests in Europe and North America.

**4. Hg budgets**
The ultimate fate of Hg deposited to the forest ecosystem may depend on its delivery and incorporation into the
forest floor. Input of THg to the forest fields included net throughfall and litterfall depositions and output pathway
from the forest ecosystem included runoff outflow and soil Hg emission back to atmosphere (St. Louis et al., 2001;
Fu et al., 2010a). A synthesis of Hg input into and output from forests, we conclude the Hg retentions in forest soils
in four subtropical forests in south China, including Tieshanping forest, Mt. Gongga, Mt. Simian and Qianyanzhou
forest (Fig. 6a). To identify how the Hg retention in the temperate forests in north China, we have also estimated the
budgets in three forest stands at Mt. Dongling in north China (Fig. 6b).
Due to no studies estimated the THg export by underground runoff in China, the underground runoff fluxes in
the four subtropical forests in south China was estimated according to the runoff amounts and THg concentrations.
The runoff amount was estimated to 25% rainfall amount (Liu et al., 2005) and THg concentration in runoff was
estimated to same as that in Tieshanping due to similar soil THg concentrations in these areas. The estimated export
fluxes by underground runoffs were ranged 6.0 to 9.9 $\mu g\ m^{-2}\ yr^{-1}$ in the four forests. Base on the budget calculation,
the THg retention (= throughfall + litterfall – runoff outflow (surface and underground) – soil-atmosphere exchange
fluxes) at the subtropical forests ranged from 26.1 to 60.4 $\mu g\ m^{-2}\ yr^{-1}$, accounted for ranging from 46.6% to 62.8%
of THg inputs (Fig. 6a). Evasion of Hg from forest soil was the dominated pathway of THg outputs from the forest
compared to runoff outflow. By comparison, the annual loading of THg to subtropical forests in China were much
higher compared to some forest catchments in Europe and North America (Larssen et al., 2008; Grigal et al., 2000).
Since atmospheric Hg distributions at subtropical areas indicated rural to suburban areas suffered heavy regional Hg
emissions from industrial and urban areas (Fu et al., 2015), we infer anthropogenic emissions caused the elevated
loading of Hg to subtropical forests in China.
In a study on Hg input at a remote temperate forest ecosystem in Mt. Changbai, northeastern China, THg
concentrations in throughfall was approximately 17 ng $L^{-1}$ (Wan et al., 2009a). The forest types at Mt. Changbai
were similar to Mt. Dongling in Beijing: mixed forest (600–1100 m a.s.l.), coniferous forest (1100–1700 m a.s.l.),



and mountain birch zone (1700–2000 m a.s.l.). Additionally, the TGM concentrations were between 1.60 ± 0.51 ng
m$^{-3}$ and 3.58 ± 1.78 ng m$^{-3}$ (Wan et al., 2009b; Fu et al., 2012), which were comparable with the concentration of
2.5 ± 0.5 ng m$^{-3}$ at Mt. Dongling (Zhou et al., 2017a). If we hypothesized the THg concentration in throughfall at
Mt. Dongling was also similar to that in Mt. Changbai and throughfall amount were estimated through the mean
interception of water-holding capacity of canopy measured by Fei et al. (2011). The estimated inputs of THg
deposition were ranged from 21.40 to 28.73 μg m$^{-2}$ yr$^{-1}$ at Mt. Dongling. As forest types in Mt. Dongling and
Changbai are similar, the forest soil types are also similar, which are both mountain brown forest soil (Wang et al.,
2013; Zhou et al., 2017a). Therefore, we also referred the Hg concentrations in runoff (5.75 ng L$^{-1}$) at Mt. Changbai
(Wang et al., 2013) and runoff volume were used a previous study in the three stands at Mt. Dongling (Fei et al.,
2011). Based on our measured THg concentrations in soil solution (9.2 ng L$^{-1}$, our unpublished data) and the amounts
of underground runoffs in the three stands (Wang et al., 2012), the export fluxes by underground runoffs were
estimated. Studies in the Chinese pine plantation, larch plantation and mixed broad-leaved forest found that the annual
emission by soil volatilization measured by dynamics chamber and were from 0.87 to 4.03 μg m$^{-2}$ yr$^{-1}$ (Zhou et al.,
2016c), and the total Hg outputs of which were 3.1, 2.5 and 9.0 μg m$^{-2}$ yr$^{-1}$, respectively. Therefore, the annual net
retention Hg from the atmosphere was 21.7 μg m$^{-2}$ yr$^{-1}$ for Chinese pine plantation, 26.2 μg m$^{-2}$ yr$^{-1}$ for larch
plantation and 12.4 μg m$^{-2}$ yr$^{-1}$ for mixed broad-leaved forest in north China. The ratios of THg retentions to the THg
inputs were much higher than these at subtropical forests (t test, p<0.05), which accounted for 57.9% to 91.3% of
THg deposition. However, it should be noted that the Hg input by throughfall and output by runoff have relative
greater uncertainties, so the Hg budget in the temperate forest is roughly estimated in the current study.

The THg retention at subtropical forests in south China were about 2.5 times these at temperate forests in north

China. If we hypothesis the total input fluxes of Hg were 20.2 and 39.2 μg m$^{-2}$ yr$^{-1}$ and output were 11.3 μg m$^{-2}$ yr$^{-1}$
(8.6 for soil emission flux, 2.7 for runoff flux) and 8.8 μg m$^{-2}$ yr$^{-1}$ (soil emission flux: 6.7, outflow flux: 2.1) for
North America and Europe, respectively, according to the average fluxes for each item, the calculated retention were
8.9 and 30.4 μg m$^{-2}$ yr$^{-1}$, respectively. The THg retention at subtropical forests was higher compared to these in
North America (3.8 to 7.9 folds) and Europe, and the retention in the temperate forest was lower compared to those
in the Europe but higher compared to those in North America (1.2 to 2.8 folds).

**5. Hg storage and risk assessment**
5.1. Hg storage in soils



Highly elevated THg contents in forest top soils were mostly likely originated from atmospheric depositions via
litterfall and throughfall, whereas very limited source was originated from geological sources (Obrist et al., 2011).
Table S1 summarizes all studies of soil Hg concentrations and pools at forests of China from the literature. However,
it is should be note that the attempts to compare soil Hg concentrations and pools with the data from each other and
some other studies are facing difficulties, because these studies either reported the amounts of THg accumulated in
different horizons or calculated THg pools stored in soil profiles of different depths, which were inconsistent with
each other.
Declining Hg concentrations with soil depth are generally observed in organic to mineral layers and did not vary
in the lower mineral soils from all the soil profiles in Chinese forests. Highest THg concentrations observed in litter
and upper soils are indicative of Hg sorption from atmospheric deposition to upper soil horizons. As organic soils are
net traps of deposited atmospheric Hg and topsoil concentrations reflect recent Hg depositions from the atmosphere,
we concluded THg concentrations from topsoil (most in the organic horizons) in the Fig. S3. The soil THg
concentration at remote forests averaged 150 ng g$^{-1}$ and the median concentration was 104 ng g$^{-1}$, ranging from 59
to 353 ng g$^{-1}$ (n = 18). The concentrations were slight higher than those observed in remote areas of North America,
which were generally less than 150 ng g$^{-1}$ for surface soils (Larssen et al., 2008; Obrist et al., 2011; Tabatchnick,
2012). The THg concentrations at rural & suburban forests were much higher than these observed at remote forests,
which ranged from 76 to 332 ng g$^{-1}$ (mean: 198 ng g$^{-1}$; median: 196 ng g$^{-1}$). This is in a good agreement with the
elevated atmospheric Hg concentrations and higher loading of Hg in at rural & suburban forests of China, which can
be proved by the significant correlation between Hg retentions and soil THg concentrations (r$^2$=0.62, p< 0.05, n=7).
Predictably, higher THg depositions and soil THg concentrations has resulted higher THg pools in forest soils. For
example, in the remote forests of Mt. Gongga and Mt. Ailao, the THg storage were up to 152.3 and 191.3 mg m$^{-2}$ in
the soil profiles of 90 and 80-cm depth, which were much higher than these in the upland forest of central Adirondack
Mountain of USA and (64 mg m$^{-2}$ in 0–90 cm depth) (Selvendiran et al., 2008) and upland forest of Steinkreuz,
Germany (19 mg m$^{-2}$ in 0–60 depth) (Schwesig and Matzner, 2000). However, THg storage in forest soils of
temperate forests and Tibet Plateau with relative lower atmospheric Hg deposition (Zhou et al., 2017a; Gong et al.,
2014), were comparable to that in North America and Europe.

5.2. Hg storage in biomass
Vegetation is known to exert significant influence the dynamics of Hg in the forest ecosystem including





atmospheric Hg input and output in the terrestrial ecosystem (Ma et al., 2016; Zhou et al., 2016a). Two studies
investigated the Hg distribution in the tissues of vegetation at the subtropical forest (Tieshanping forest, Zhou et al.,
2016) and temperate forest (Mt. Dongling, Zhou et al., 2017a) and showed that the THg concentration followed the
order of Oa > Oe > Oi > litterfall > leaf/needle > root > bark > branch > bole wood for each species. Highest THg
concentrations are observed in the O horizons compared to THg in the other biomass, because organic matter was
enhanced during natural processes of litterfall decomposition and transformation, in which organic matter binding
Hg compounds are usually more stabilized via complexing, humification and adsorption to clay minerals (Demers et
al., 2007; Pokharel and Obrist, 2011; Zhou et al., 2017a). Sequentially, relative higher THg was observed in the
litterfall and leaf due to canopy leaf can effectively capture Hg in atmosphere, which can uptake Hg by stomata (Fu
et al., 2015).
Root is contacted with mineral soil directly, likely to higher concentration than that of aboveground wood (Grigal,
2003). THg concentrations in roots of Norway spruce in southern Sweden were 40 ng g$^{-1}$ (Munthe et al., 1998), which
was much lower than that in the root of Masson pine in southwestern China (71 ng g$^{-1}$, Zhou et al., 2016a) due to
large THg loading in this area. Mass of tree roots is about one-fifth that of aboveground material (Wharton and
Griffith, 1993; Whittaker and Marks, 1975) and combined with high THg concentration, roots may store much higher
THg biomass compared to other plant components. However, data are rare for these pools of THg at forests. Only
Zhou et al. (2016) estimated the THg pools in roots that accounted for about 34% of the overstory THg pools. Bole
wood had the largest biomass of vegetation in the forest, but lowest THg concentrations were observed. A previous
study suggested that the source of the THg in wood was translocated from foliage (Barghigiani et al., 1991).
Concentrations of Hg were positively correlated in 11 pairs of leaf and adjacent bole wood samples of different tree
species at forests of China (Fig. S4). It is reasonable for their correlation because leaf and bole wood are both exposed,
one directly and the other indirectly to the same atmospheric pool of Hg. Grigal (2003) suggested that THg in bark
is probably from long-term dry deposition, and they summarized 15 pairs of bark and adjacent wood-only samples
and found significant correlations. However, no significant correlation was observed between THg concentrations in
bark and bole wood or leaf, probably due to that the THg accumulation rates were differed in the barks of different
tree species.
THg concentrations of each component at the suburban forest of Tieshanping at subtropical zone was much
higher than those at the remote forest of Mt. Dongling at temperate zone. Accordingly, much higher THg pool of
103.5 mg m$^{-2}$ showed in suburban forest of Tieshanping than that of 7.3–10.8 mg m$^{-2}$ in remote forest of Mt. Dongling



(Fig. S5). The THg pools in North America were much lower than those at subtropical forest of China and comparable
to those at temperate forest of China (Friedli et al., 2007; Obrist et al., 2009; Richardson et al., 2013). Nonetheless,
soil THg pools accounted for over 90% of the total ecosystem Hg pools forests around the world. For example, over
97% and 99% of the THg resides in soil layers (0–40 cm) at Mt. Dongling and Tieshanping forest in China; more
than 99% of the THg pool were stored in the soil depth of top 60 cm at the coniferous and deciduous upland forest
in Vermont, USA (Richardson and Friedland, 2015); THg pools at upland forest in Sierra Nevada, showed soil of top
40 cm constituted over 94% of the total ecosystem Hg storage (Obrist et al., 2009); THg pools in the soils exceed
more than 90% of the total ecosystem Hg pools at Sierra Nevada forest (Engle et al., 2006; Obrist et al., 2009); and
THg resided in organic soils accounted from 93 to 97% of ecosystem THg at two subtropical forest stands in Canada
(Friedli et al., 2007).

5.3. Risk assessment

The studies summarized in this review showed significant inputs and retention of Hg in forest ecosystems in

China. The apparent accumulation and storage of THg may present an important ecological risk. Firstly, the Hg in
forest soil could be re-emitted back to the atmosphere. Organic matter has a high binding ability of Hg in forest
surface soils, but the Hg bonded organic carbon would probably be released to the environment as the decomposition
of organic matter occurs. Studies on climate change showed that the accelerated global warming would accelerate
the decomposition of organic carbon (Schimel et al., 1994), which could probably accelerate Hg emission from soil
(Obrist, 2007; Fu et al., 2010a). Additionally, the increasing of global temperature would aggravate the occurrence
of potential fires and causing large pulses of Hg to the global atmospheric pool (Zhou et al., 2016a, 2017a). The
average THg emission from forest wildfires was 0.78 t yr$^{-1}$ during the first decade of this century in China (Chen et
al., 2013), which was accounting for about 12.8% of total Hg emissions from biomass burning. Zhou et al. (2016a)
estimated the THg emission from the subtropical forest of Tieshanping was about 0.82 mg m$^{-2}$, which was lower than
the mean value of 1.22 mg m$^{-2}$ (range: 0.68–1.70 mg m$^{-2}$) in the temperate forest of Mt. Dongling (Zhou et al., 2017a).
In contrast, the THg pools in the fuel biomass were much higher at the subtropical forest compared the temperate
forest as we showed in the above section. Therefore, it should be noted that THg emission rate from different plant
components and soil layers differed greatly due to combustion completeness that is defined as the ratio of THg
concentration loss by wildfire to THg concentration before burn (Melendez-Perez et al., 2014). Due to the large
amount of THg retention in Chinese forests, a hectare of forest combustion equals about from 104.4 to 261.5 t coal



combustion in China (Zhou et al., 2016a, 2017a).
Secondly, the Hg retention in the forest soils would accumulate through food webs, threatening the balance of
forest ecosystems (Rimmer, 2010). However, the relevant studies in China were rare. Many studies showed that
mushroom had high accumulation ability of THg and MeHg from substrate (like soil, litter and wood) and strong
translocation to the fruiting bodies (Fischer, et al., 1995; Árvay et al., 2014; Falandysz et al., 2015a, b, 2016; Ostos
et al., 2015). Studies in southwestern China showed that THg concentrations in the Fungi *Boletus* species and genus
*Leccinum* species were up to 3500–4800 ng g$^{-1}$ (mean 42000 ng g$^{-1}$) and 4900–22000 ng g$^{-1}$ (10900 ng g$^{-1}$) dry
matter, respectively (Falandysz et al., 2015a, b). Similarly, a study in Poland also showed efficient accumulation of
THg in the *Leccinum* mushrooms, but the average Hg concentrations being an order of magnitude lower because of
lower concentrations of THg in surface forest soil of Poland. Although some lowly cumulative species of mushroom
were observed in the subtropical forests (Kojta et al., 2015; Wiejak et al., 2014), mushroom is an important food item
in southwestern China, and high rates of consumption can deliver relatively high doses of Hg to local human beings
(Kojta et al., 2015; Falandysz et al., 2015a, b, 2016). If according to the value of the provisionally tolerable weekly
intake (PTWI) or the reference dose (RfD), the most edible mushrooms from Yunnan provide a high dose of Hg when
consumed at a rate higher than 300 g per week, which will post a higher health risks to consumers (Falandysz et al.,

2016).

Additionally, the ecological stress to forest insect were investigated in an suburban forest (Tieshangping) in
China, which showed that insect living in the soil has two to three orders of magnitude higher THg accumulation
than that living on the plant due to large Hg pools in the forest soils (Zhou et al., 2016a). Although animals in the
high position of the food chain were not studied in forest of China, Rimmer et al. (2010) showed that food web
reflected the transfer of Hg from lower to higher trophic levels with a resulting increase in Hg burden. Therefore, we
can infer that Hg will be seriously bioaccumulated along the food chain and pose risk to the local creatures by
physiological toxicity.

**6. Environmental implication and research needs**
The large THg retention of in the forest ecosystem suggested strong adsorption and absorption of Hg by
vegetation that was underestimated by global modeling of previous studies. If we roughly estimated the THg
deposition at forests of China using the average THg depositions (92.45 μg m$^{-2}$ yr$^{-1}$) by present studies and the forest
area (2.08×10$^{12}$ m$^2$) in 2015, the THg deposition would be 192.3 t yr$^{-1}$ in forest areas of China. GEOS-Chem model



estimates the mean dry deposition of 12.3 mg m$^{-2}$ yr$^{-1}$, which converted to the total Hg deposition in China is <121.0
t yr$^{-1}$ (Wang et al., 2014). Given that more than 80% of the THg deposition was from dry deposition, the THg dry
deposition was 153.8 t yr$^{-1}$ in forest ecosystems of China, which is even higher than the total Hg deposition in the
whole mainland China. Therefore, a large underestimation compared to the observation-based estimate just from
forest areas of China in this study. Therefore, future model studies should consider the THg dry deposition in forested
areas individually.

Hg sequestrated in forest litters and surface soil by legacy Hg retention can be quickly volatilized to the

atmosphere by soil-atmosphere exchanges. Recent global Hg models suggested that soils not only act as net sinks but
also as net sources for atmospheric Hg in global Hg cycling (Amos et al., 2013), and the role of forest ecosystems as
atmospheric Hg sink or a source are existing confliction (Lindberg et al., 1991, 1998; Pirrone et al., 2010; Gustin et
al., 2008). Using the global database of terrestrial surface−atmosphere fluxes, forest ecosystems appear a net
deposition of 59 t yr$^{-1}$, but the estimation existed large uncertainties and ranged (37.5th−62.5th percentiles) from a
deposition of 727 t yr$^{-1}$ to an emission of 703 t yr$^{-1}$ (Agnan et al., 2016). Base on the field observations of THg
retention in Chinese forests, the THg retention in forest soils was 69 t yr$^{-1}$ just in China, which was much higher than
the global data of 59 t yr$^{-1}$ (Agnan et al., 2016). Such difference is mainly resulted from the variation of reported
atmospheric Hg uptake by foliage and the limited geospatial representation of available data (Wang et al, 2016; Zhu
et al., 2016; Agnan et al., 2016). Thus, more studies should be conducted to character the whole-ecosystem fluxes
and to question to what degree the ecosystems are net sinks or sources of atmospheric Hg.

To better assess the role of forest ecosystems in the global Hg cycling, it is also essential to understand the THg

pools in the branches, stems and roots that can be translated from the atmosphere by the foliage uptake. A previous
study estimated that approximately 139 t yr$^{-1}$ Hg was stored in bole woods (Obrist et al., 2007). However, there is no
study exactly quantifying the amount of Hg translocation after Hg uptake by leaves, and the THg storage in biomass
are scarce and need more data. Further studies concerning the transformation and migration processes after vegetation
uptake will benefit to constrain atmospheric Hg sink in forest ecosystems.

In addition, the large "active" soil pool at forests is a potential short-term and long-term source of THg and

MeHg to downstream aquatic ecosystems (Selvendiran et al., 2008; Ma et al., 2015). However, there is no study
reporting the accumulation of THg and MeHg in aquatic ecosystem after output from the forest ecosystem. The
processes of Hg methylation, transformation and translocation may be different from those in North America and
Europe because of the larger Hg deposition and storage in China, which requires further investigation when more



data become available.

**7. Conclusions**

As the largest energy consumer and anthropogenic Hg emission in the world, much attention has been paid to

characterize the behavior of Hg in China. Forests are regarded as large pools of Hg in the global Hg cycle. In this
paper, an integrated review of the knowledge reported in peer-reviewed literature is provided. Hg deposition and
pools have been found to be substantially elevated in both remote, rural & suburban forests of China compared to
those observed in North America and Europe. A strong spatial variation in Hg pools was observed, with high storage
related to regional atmospheric Hg concentrations in southern China. The large Hg storage in the forests pose a
serious threat for large pluses to the atmospheric Hg during accelerated organic matter decomposition and potential
wildfires, and additional ecological stress to forest animals. However, very few studies are attempted to study the
ecological risk of Hg in the forest ecosystem in China, which are suffering highly Hg depositions.

The forests play important roles in the geochemical cycles of Hg in China. According to the budget calculation,

the THg retention ranged from 26.1 to 60.4 $\mu g\ m^{-2}\ yr^{-1}$ at the subtropical forests in southern China, accounted for
ranging from 46.6% to 62.8% of THg inputs, and ranged from 12.4 to 26.2 $\mu g\ m^{-2}\ yr^{-1}$ at the temperate forests in
northern China. The Hg retention and storage pools in at the subtropical forests were much higher than those in North
America, but those in the temperate forests were comparable to Europe and North America. The result of the current
review may answer the question to what degree the ecosystems are net sinks or sources of atmospheric Hg in China.
However, further studies are needed to accurately quantify Hg budgets and retentions of Hg in different forests
ecosystems in China, as well as the atmospheric Hg budget in China.

**Acknowledgments**

This research was funded by the National Science and Technology Support Plan (2015BAD05B01), the National

Basic Research Program of China (No. 2013CB934302 and 2013CB430002) and National Natural Science
Foundation of China (41701361 and 4157146). The anonymous reviewers are acknowledged for providing insightful
comments and suggestions.. The anonymous reviewers are acknowledged for providing insightful comments and
suggestions.






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





**Table 1.** Hg concentrations (ng L$^{-1}$ or ng g$^{-1}$) and deposition fluxes (µg m$^{-2}$ yr$^{-1}$) in precipitation, throughfall, and
litterfall in China.

| Site | Forest type | Altitude (m a.s.l) | Location Type | Study period | Samples | Concentration | | Deposition flux | | References |
|---|---|---|---|---|---|---|---|---|---|---|
| | | | | | | THg | MeHg | THg | MeHg | |
| Mt. Ailao, Yunnan | Subtropical evergreen broadleaf mixed | 2500 | Remote | 06/2011–05/2012 | Precipitation | 3.0 | 0.08 | 5.4 | 0.14 | Zhou et al., 2013a, b |
| | | | | | Litterfall | 54.0 | 0.28 | 71.2 | 0.36 | |
| Mt. Ailao, Yunnan | Subtropical evergreen broadleaf mixed | 2500 | Remote | 2011–2014 | Precipitation | 4.9 | | 4.9 | | Wang et al., 2016 |
| | | | | | Litterfall | 43–66 | | 75.0 | | |
| | | | | | Throughfall | 22.9 | | 20–30 | | |
| Mt. Leigong, Guizhou | Subtropical deciduous broadleaf mixed | 2178 | Remote | 05/2008–05/2009 | Precipitation | 4.0 | 0.04 | 6.1 | 0.06 | Fu et al., 2010a |
| | | | | | Throughfall | 8.9 | 0.1 | 10.5 | 0.12 | |
| | | | | | Litterfall | 91.0 | 0.48 | 39.5 | 0.28 | |
| Mt. Leigong, Guizhou | Subtropical deciduous broadleaf mixed | 1680 | Remote | 03/2005–02/2006 | Precipitation | 12.9 | | 16.8 | | Wang et al., 2009 |
| | | | | | Throughfall | 36.7 | | 41.2 | | |
| | | | | | Litterfall | 135.1 | | 78.3 | | |
| Mt. Damei, Zhejiang | Subtropical deciduous broadleaf mixed | 550 | Remote | 08/2012–08/2014 | Precipitation | 3.7 | | 6.0 | | Fu et al., 2016 |
| | | | | 08/2012–07/2013 | Litterfall | 42.3 | | 23.1 | | |
| Mt. Gongga, Sichuan | Subtropical evergreen broadleaf | 1640 | Remote | 01–12/2006 | Precipitation∗ | 9.9 | | 9.1 | | Fu et al., 2008b |
| Mt. Gongga, Sichuan | Subtropical evergreen broadleaf | 3000 | Remote | 05/2005–04/2006 | Precipitation∗ | 14.2 | 0.16 | 26.1 | 0.30 | Fu et al., 2010b |
| | | | | | Throughfall | 40.2 | 0.3 | 57.1 | 0.43 | |
| | | | | | Litterfall | 35.7 | | 35.5 | | |
| Mt. Changbai, Jilin | Temperate broadleaf and pine mixed | 750 | Remote | 08/2005–07/2006 | Precipitation∗ | 13.4 | | 8.4 | | Wan et al., 2009a |
| | | | | | Throughfall | 9.0 | | 24.9 | | |
| Mt. Changbai, Jilin | Temperate broadleaf and pine mixed | 736 | Remote | 08/2011–08/2014 | Precipitation | 7.4 | | 5.6 | | Fu et al., 2016 |
| | | | | | Litterfall | 47 | | 22.8 | | |
| Mt. Dongling, Beijing | Temperate Chinese pine evergreen | 1100 | Remote | 09–11/2015 | Litterfall | 39.8 | | 15.8 | | Zhou et al., 2017a |
| | Temperate larch deciduous | | | | Litterfall | 63.3 | | 19.6 | | |
| | Temperate oak deciduous | | | | Litterfall | 46.5 | | 14.1 | | |
| | Temperate mixed deciduous | | | | Litterfall | 45.3 | | 12.9 | | |
| Linzhi, Tibetan | Subtropical evergreen coniferous | 3200 | Remote | 8/2008 | Litterfall | 12.6 | | 4.2 | | Gong et al., 2014 |
| China (22 sites) | Suburban evergreen broadleaf | | Suburban | | Litterfall | 50.8 | | 17.9 | | Niu et al., 2011 |
| | Suburban deciduous broadleaf | | | | Litterfall | 25.8 | | 8.73 | | |
| Tieshanping, Chongqing | Subtropical evergreen coniferous | 500 | Suburban | 03/2005–03/2006 | Precipitation | 32.3 | | 29.0 | | Wang et al., 2009 |
| | | | | | Throughfall | 69.7 | | 71.3 | | |
| | | | | | Litterfall | 105 | | 220 | | |
| Tieshanping, Chongqing | Subtropical evergreen coniferous | 500 | Suburban | 2010–2011 | Throughfall | 69 | | 67.5 | | Luo et al., 2015a |
| | | | | | Litterfall | 115 | | 22.3 | | |
| Tieshanping, | Subtropical evergreen coniferous | 500 | Suburban | 04/2014– | Litterfall | 85 | 0.21 | 40.51 | 0.10 | Zhou et al., |



| | | | | | | | | | | |
|---|---|---|---|---|---|---|---|---|---|---|
| Chongqing | Subtropical evergreen broadleaf | | | 03/2015 | Litterfall | 89 | 0.23 | 90.85 | 0.34 | 2016c |
| Mt. Jinyun, Chongqing | Subtropical evergreen broadleaf | 900 | Rural | 03/2012– 02/2013 | Precipitation | 11.9 | 0.20 | 15.9 | 0.26 | Ma et al., 2015 |
| | | | | | Throughfall | 20.1 | 0.55 | 21.8 | 0.60 | |
| | | | | | Litterfall | 104.5 | 0.84 | 43.5 | 0.27 | |
| Mt. Simian, Chongqing | Subtropical evergreen broad-leaf | 1394 | Rural | 03/2012– 02/ 2013 | Precipitation | 10.9 | 0.24 | 15.45 | 0.36 | Ma et al., 2016 |
| | | | | | Throughfall | 24.04 | 0.33 | 32.17 | 0.45 | |
| | | | | | Litterfall | 106.7 | 0.79 | 42.89 | 0.32 | |
| Qianyanzhou, Jiangxi | Subtropical evergreen coniferous | 60 | Rural | 11/2013– 12/2014 | Precipitation | 23 | | 14.4 | | Luo et al., 2015a |
| | | | | | Throughfall | 42 | | 34.6 | | |
| | | | | | Litterfall | 42.9 | | 21.4 | | |
| Huitong, Hunan | Subtropical evergreen coniferous | 335 | Rural | 4/2013– 12/2014 | Precipitation | 12.5 | | 15.9 | | Luo et al., 2015a |
| | | | | | Throughfall | 29.9 | | 27.8 | | |
| | | | | | Litterfall | 176.1 | | 33.6 | | |
| Luchonguan, Guizhou | subtropical broad-leaf and coniferous | 1360 | Urban | 01/2005– 01/2006 | Throughfall | 43.6 | | 49.0 | | Wang et al., 2009 |







**Table 2.** Hg concentrations (ng L$^{-1}$ or ng g$^{-1}$) and export fluxes (µg m$^{-2}$ yr$^{-1}$) in stream water/runoff in China.

| Site | Forest type | Altitude (m a.s.l) | Location Type | Study period | THg concentration | THg export flux | References |
|---|---|---|---|---|---|---|---|
| Northeast China | Temperate evergreen/deciduous coniferous and broadleaf | 442 ± 324 | Remote and rural | | 17.2±11.0 | | Luo et al. 2014 |
| South China | Subtropical evergreen conifers/mixed broad-leaved | 548 ± 295 | Remote and rural | | 6.2 ±6.4 | | Luo et al. 2014 |
| Mt. Leigong, Guizhou | Subtropical deciduous broadleaf mixed forest | 1680 | Remote | 03/2005–02/2006 | 4.3±2.5 | 3.0 | Wang et al., 2009 |
| Mt. Changbai, Jilin | Temperate broadleaf and pine mixed | 750 | Remote | 04/2009, 09/2009 | 5.5 ± 4.1 | | Wang et al., 2013 |
| Tieshanping, Chongqing | Subtropical evergreen coniferous | 500 | Suburban | 03/2005–03/2006 | 6.2 ±3.5 | 3.5 | Wang et al., 2009 |
| Tieshanping, Chongqing | Subtropical evergreen coniferous | 500 | Suburban | 04/2014 | 3.1 ± 1.2 | | Zhou et al., 2015a |
| Luchongguan, Guizhou | Subtropical broad leave-coniferous mixed subtropical | 1360 | Urban | 01/2005–01/2006 | 8.9± 6.7 | 4.5 | Wang et al., 2009 |
| Mt. Gongga, Sichuan | Subtropical evergreen broadleaf | 3000 | Remote | 05/2005–04/2006 | 3.5±0.9 | 8.6 | Fu et al., 2010a |
| Mt. Simian, Chongqing | Subtropical evergreen broad-leaf | 1394 | Rural | 03/2012–02/ 2013 | 3.9 ±2.0 | 7.23 | Ma et al., 2016 |
| Huitong, Hunan | Subtropical evergreen coniferous | 335 | Rural | 4/2013–12/2014 | 4.9 | 2.03 | Luo et al., 2015a |
| Qianyanzhou, Jiangxi | Subtropical evergreen coniferous | 60 | Rural | 11/2013–12/2014 | 2.3 | | Luo et al., 2015a |





**Table 3.** Soil-atmosphere Hg exchange fluxes (ng m$^{-2}$ hr$^{-1}$), soil Hg concentrations and surface TGM
concentrations (ng m$^{-3}$) in atmosphere in forested areas of China and other regions.

| Locations | Forest Type | Altitude | Location Type | Study period | Soil Hg | Surface TGM | Flux | References |
|---|---|---|---|---|---|---|---|---|
| Mt. Dongling, Beijing (Temperate) | Chinese Pine | 1050 | Remote | 07/2015–05/2016 | 88 | 2.2 ±1 | 0.01 ±2.6 | Zhou et al. 2016c |
| | Larch | 1020 | Remote | 07/2015–05/2016 | 69 | 2.3 ±1 | 0.12 ±1.28 | Zhou et al. 2016c |
| | Mixed broadleaf forest | 1250 | Remote | 07/2015–05/2016 | 54 | 2.4 ±1 | 0.46 ±1.36 | Zhou et al. 2016c |
| | Wetland | 1150 | Remote | 07/2015–05/2016 | 154 | 2.5 ±1.1 | 3. 6 ±6.8 | Zhou et al. 2016c |
| Mount Gongga, Sichuan (Subtropical) | Shrub | 2350 | Remote | 21– 22/08/2006 | 90 | 3.6 ±1.3 | 6.6 ±4.2 | Fu et al., 2008 |
| | Broadleaf Forest | 1220 | Remote | 27– 29/08/2006 | 60 | 3.7 ±0.5 | 5.7 ±4.7 | Fu et al., 2008 |
| | Broadleaf Forest | 1650 | Remote | 17–18/08/ 2006 | 110 | 2.3 ±0.4 | 9.3 ±4.3 | Fu et al., 2008 |
| | Broadleaf Forest | 2140 | Remote | 19– 21/08/ 2006 | 180 | 2.3 ±0.3 | 7.7 ±3.4 | Fu et al., 2008 |
| | Broadleaf Forest | 2500 | Remote | 30–31/08/2007 | 160 | 2.0 ±0.6 | 0.5 ±1.8 | Fu et al., 2008 |
| | Pine forest | 3050 | Remote | 31/08–1/09/2008 | 80 | 1.6 ±0.6 | 2.9 ±2 | Fu et al., 2008 |
| Mount Gongga, Sichuan (Subtropical) | Evergreen broadleaf | 3000 | Remote | 17/8/2006– 1/9/2013 | | 4.7 | 1.6 | Fu et al., 2010a |
| Mt. Simian, Chongqing (Subtropical) | Evergreen broadleaf | 1394 | Rural | 19/8/2003 | 174 | 19.9 ±8.6 | 7.7 ±3.9 | Wang et al., 2006 |
| Mt. Jinyun, Chongqing (Subtropical) | Evergreen broadleaf | 900 | Rural | 9/15/2003 | 137 | 9.9 ±1.8 | 3.4 ±1.5 | Wang et al., 2006 |
| Mt. Gele, Chongqing (Subtropical) | Evergreen broadleaf | 600 | Rural | 6/1/2003 | 196 | 14.1 ±3 | 8.4 ±2.5 | Wang et al., 2006 |
| Mt. Jinyun, Chongqing (Subtropical) | Mixed broadleaf-conifer | 900 | Rural | 4/2012–1/2013 | | | 14.2 ±10.9 | Ma et al., 2015 |
| | Shrub | 900 | Rural | 5/2012–1/2013 | | | 16.9 ±13.3 | Ma et al., 2015 |
| | Bamboo | 900 | Rural | 4/2012–2/2013 | | | 17.8 ±14.2 | Ma et al., 2015 |
| | Deciduous broadleaf | 900 | Rural | 4/2012–2/2013 | | | 12.2 ±10.7 | Ma et al., 2015 |
| Mt. Jinyun, Chongqing (Subtropical) | Mixed broadleaf-conifer | 900 | Rural | 4/2011–3/2012 | 140 | | 14.2 ±6.7 | Ma et al., 2014 |
| Mt. Simian, Chongqing (Subtropical) | Deciduous broadleaf | 1394 | Rural | 3/2012–2/2013 | 161 | | 12.12 ±10.7 | Ma et al., 2016 |
| Qianyanzhou, Jiangxi (Subtropical) | Evergreen coniferous | 60 | Rural | 11/2013–12/2014 | 101 | 3.6 | 2.1 | Luo et al., 2015a |
| Tieshanping, Chongqing (Subtropical) | Masson pine | 500 | Suburban | 03/2014–01/2015 | 219 | 3.6 ±1.3 | 2.76 ±3.85 | Zhou et al. 2016c |
| | Masson pine | 500 | Suburban | 03/2014–01/2015 | 264 | 3.8 ±1.3 | 3.52 ±4.18 | Zhou et al. 2016c |
| | Camphor | 500 | Suburban | 03/2014–01/2015 | 156 | 3.3 ±1.4 | 0.18 ±2.24 | Zhou et al. 2016c |
| | Wetland | 500 | Suburban | 03/2014–01/2015 | 96 | 4.9 ±2 | –0.8 ±5.05 | Zhou et al. 2016c |
| Tieshanping, Chongqing (Subtropical) | Masson pine | 500 | Suburban | 09/2012–07/2013 | 294 | 5.2 ±2 | 0.3 ±0.8 | Du et al., 2014 |
| Nanhu, Changchun (Temperate) | Temperate | | Urban | | 143 | | 7.6 | Fang et al., 2003 |
| Jingyuetan, Changchun (Temperate) | Temperate | | Urban | | 136 | | 3.3 | Fang et al., 2003 |





| Zhuzhou, (Subtropical) | Hunan | Mixed conifer | broadleaf- | Contamin ated | 09/2012–03/2014 | 3190 | 13.8 | 15.3±2.8 | Du et al., 2014 |
|---|---|---|---|---|---|---|---|---|---|





**Figure captions:**

**Fig. 1.** Contributions to the Hg input fluxes ($\mu g\ m^{-2}\ yr^{-1}$) to forests from precipitation, throughfall, litterfall and
total inputs (throughfall + litterfall) in China.
**Fig. 2.** Relationship analysis between the GEM or TGM concentrations verses the litterfall Hg concentrations
for field trap measurements.
**Fig. 3.** Correlations between litterfall deposition fluxes of Hg and (a) mass-weighted mean (MWM) Hg
concentrations in litterfall, (b) litterfall biomass.
**Fig. 4.** Box chart for Hg inputs to forest ecosystems in China, Europe and North America.
**Fig. 5.** Box chart for soil-atmosphere Hg exchange fluxes in deciduous and evergreen forest ecosystems in China
(CHI, including four seasons), North America (NA), Europe (Eur) and Brazil (Bra).
**Fig. 6.** Total mercury budgets ($\mu g\ m^{-2}\ yr^{-1}$) at the three temperate forest stands of Mt. Dongling (a) and four
subtropical forests of Tieshanping, Qianyanzhou, Mt. Gongga and Mt. Simian forests.





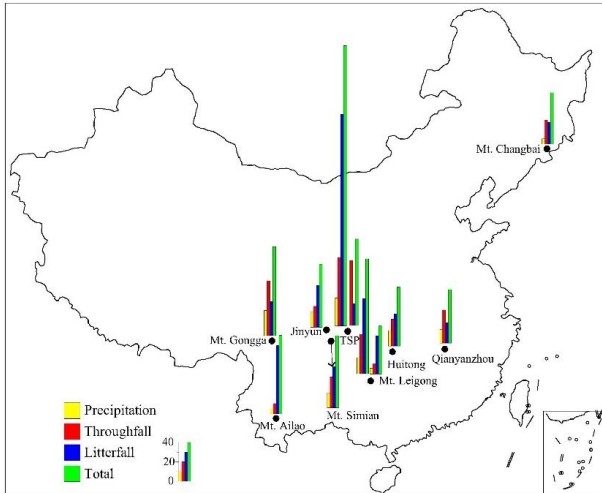

**Fig. 1.** Contributions to the Hg input fluxes ($\mu$g m$^{-2}$ yr$^{-1}$) to forests from precipitation, throughfall, litterfall and total inputs (throughfall + litterfall) in China. Mt. Ailao, Mt. Leigong, Mt. Gongga and Mt. Changbai are regarded as remote sites and Mt. Jinyun, Mt. Simian, Qianyanzhou, Huitong and Tieshanping (TSP) are regarded as suburban and rural sites.



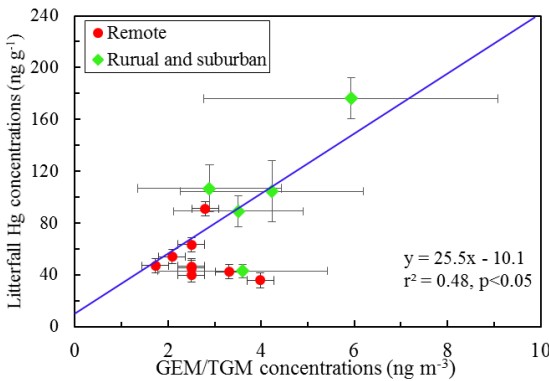

925

**Fig. 2.** Relationship analysis between the GEM or TGM concentrations verses the litterfall Hg concentrations for
field trap measurements. Data were from Mt. Ailao (Zhou et al., 2013a; Zhang et al., 2015), Mt. Leigong (Fu et al.,
2010a, b), Mt. Damei (Lang et al., 2015; Yu et al., 2015), Mt. Gongga (Fu et al., 2008a, b), Mt. Changbai (Fu et al.,
2016, 2014), Mt. Dongling (Zhou et al., 2017a), Mt. Jinyun (Ma et al., 2015), Mt. Simian (Ma et al., 2016),
Qianyanzhou (Luo et al., 2015), Huitong (Luo et al., 2015) and Tieshanping (Zhou et al., 2016a).



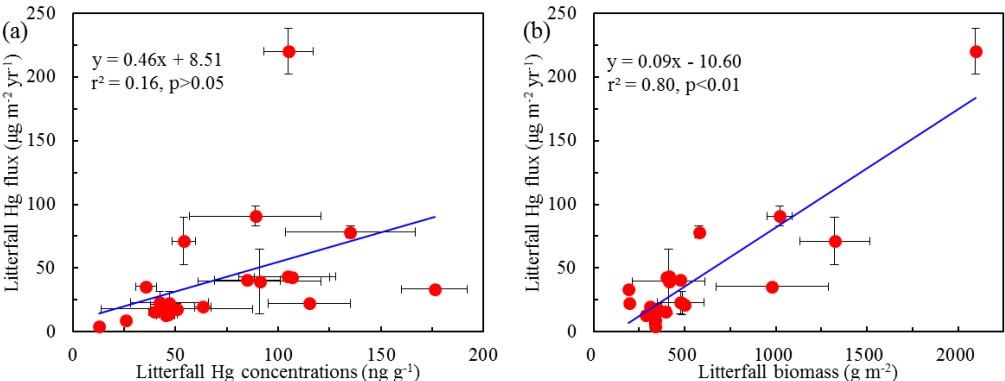

933

**Fig. 3**. Correlations between litterfall deposition fluxes of Hg and (a) mass-weighted mean (MWM) Hg
concentrations in litterfall, (b) litterfall biomass. Data are from Zhou et al., 2013a, 2016a, 2017a; Fu et al., 2010a, b,
2016; Luo et al., 2015a, b; Wang et al., 2009; Gong et al., 2014; Niu et al., 2011; Ma et al., 2015, 2016; Luo et al.,
2015a.



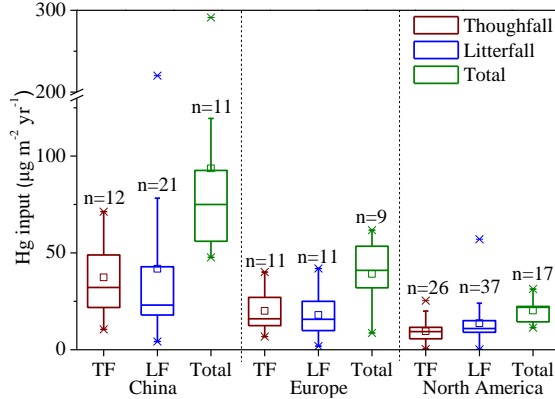

**Fig. 4.** Box chart for Hg inputs to forest ecosystems in China, Europe and North America. "TF" is the throughfall; "LF" is the litterfall; "Total" is the total Hg input (throughfall + litterfall) to the forest ecosystem. Data are from Hultberg et al., 1995; Iverfeldt et al.,1991; Larssen et al., 2008; Lee et al., 2000; Munthe et al., 1995, 1998; Schwesig and Matzner, 2000, 2001, 2003; Xiao et al., 1998; Blackwell and Driscoll, 2015a, b; Bushey et al., 2008; Choi et al., 2008; Demers et al., 2007; Kalicin et al., 2008; Kolka 1999; Grigal et al., 2000; Lindberg et al., 1994, 1996; Fisher and Wolfe, 2012; Friedli et al., 2007; Rea et al., 1996, 2001, 2002; Johnson, 2002, Johnson, et al., 2007; Nelson et al., 2007; St. Louis et al., 2001; Graydon et al., 2008; Juillerat et al., 2012; Obrist et al., 2012; Richardson and Friedland, 2015; Risch et al., 2012; Sheehan et al., 2006; Selvendiran et al., 2008; Zhou et al., 2013a, 2016c, 2017a; Zhang et al., 2015; Lang et al., 2015; Yu et al., 2015; Fu et al., 2008a, b, 2010a, b, 2016, 2014; Ma et al., 2015, 2016; Luo et al., 2015.



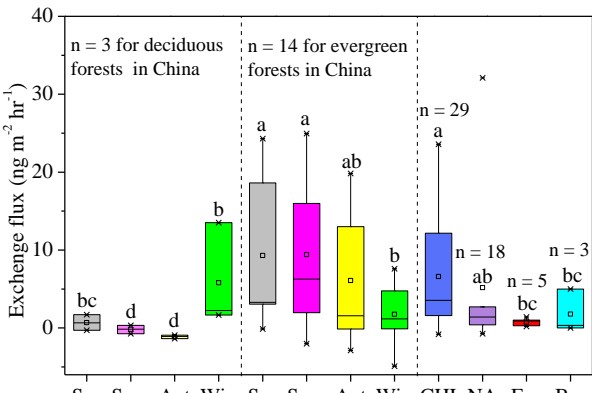

**Fig. 5.** Box chart for soil-atmosphere Hg exchange fluxes in deciduous and evergreen forest ecosystems in China (CHI, including four seasons), North America (NA), Europe (Eur) and Brazil (Bra). "Spr" is spring; "Sum" is summer; "Aut" is autumn; "Win" is winter. The post hoc tests (Tukey's HSD) were performed at 5% significance level. Data for deciduous forest in China are from Zhou et al. 2016c; for evergreen forests are from Du et al., 2014, Fu et al., 2008c, 2010a, Wang et al., 2006, Ma et al., 2014, Ma et al., 2016, Luo et al. (2015a), Fang et al., 2003; Zhou et al. 2016c; for North America are from Ericksen et al., 2006, Hartman et al., 2009, Carpi and Lindberg, 1998, Kuiken et al. 2008a, b, Lee et al. 2000, Lindberg et al. 2002, 1998, Poissant et al. 2004, Poissant and Casimir, 1998, Carpi et al., 2014, Choi and Holsen, 2009, Zhang et al., 2001, Schroeder et al., 1989; for Europe are from Xiao et al. 1991, Kyllönen et al., 2012, Lindberg et al. 1998; from Brazil are from Almeida et al., 2009; Carpi et al., 2014; Magarelli and Fostier, 2005.

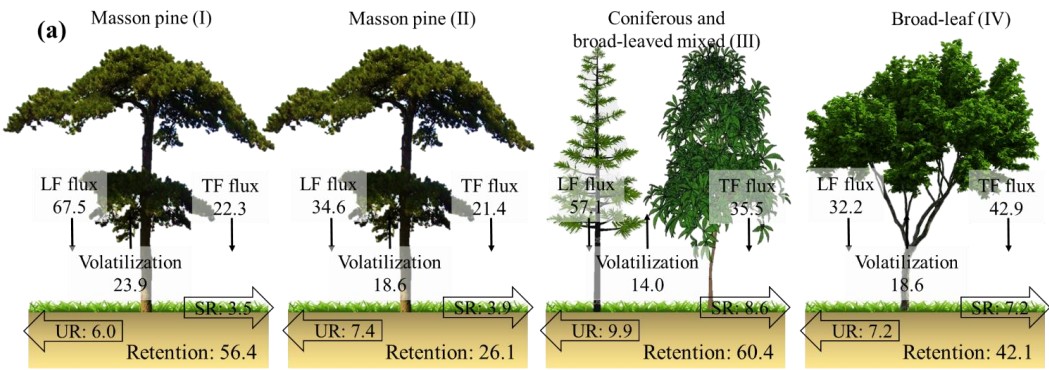

965

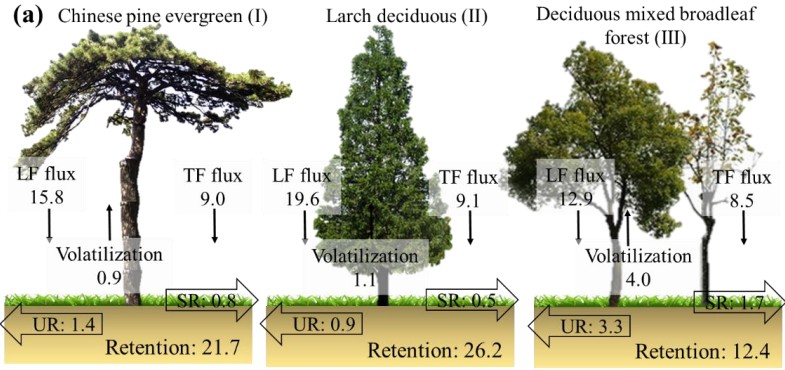

966

**Fig. 6.** Total mercury budgets ($\mu$g m$^{-2}$ yr$^{-1}$) at the four subtropical forests of Tieshanping (I), Qianyanzhou (II), Mt. Gongga (III) and Mt. Simian forests (IV) (a) and three temperate forest stands of Mt. Dongling (I-III) (b). LF, TF, SR and UR represent litterfall, throughfall, surface runoff and underground runoff fluxes, respectively. Data are from Zhou et al. (2016a, c), Luo et al. (2015b), Wang et al. (2009), Luo et al. (2015a), Fu et al. (2010a), Ma et al., 2016.