# Peer review of "Mercury fluxes, budgets and pools in forest ecosystems of China: A critical review"

_Atmospheric Chemistry and Physics, 2017_

## Referee Comment (RC1) · Anonymous Referee #2 · 16 May 2018

This manuscript reviewed the mercury fluxes, budgets and pools in forest ecosystems in China, however, the authors did not do a good systemic summary for all the current research. For example, the underground flow was not considered into the budget; There are numerous typographical errors and poor sentence structure throughout he paper. Tieshanping showed a high Hg pool, why? , due to its location or being influenced by human activities. Line 29, Chinses?? Line 43, change are to is Line 50, change sinks to sink Line 120, change plays to play Line 136, change resulted to result; which is closed to the large Hg mine of Wanshan, I think your mean is that the litterfall Hg concentration at Leigong is closed to the Hg concentration of litterfall in Wanshan?? If like this, please reorganize this sentence. Line 152, change we to it Line 154, change was to were Line 157, change ranging to range Line 209, change has to have Line 214,

change depends to depend Line 223, only one sites?? Line 239, change were to was Line 247, change showed to show Line 277, humidity?? Line 287,change were to was Line 306, add were after forests Line 439, change was to were

---

## Referee Comment (RC2) · Anonymous Referee #1 · 9 Jul 2018

The manuscript by Zhou et al. attempts to "provide a better understanding of current knowledge with respect to forest Hg in China and quantify the forest act as net sinks or sources of GEM" and discuss the ecological risk of Hg accumulation in forest ecosystems. Although the authors provide a reasonable summary regarding Hg concentrations in streams with associated "export" fluxes (this is perhaps an ill-defined term by the authors since there is no sufficient evidence that the Hg measured in streams represent "removal" or "export" of Hg from the environmental systems under discussion) and present simplified graphic illustrations for Hg mass balance in various type of forest ecosystems, the manuscript has major deficiencies that do not meet the publication standards of Atmospheric Chemistry and Physics.

1. First of all, I am not sure if the review paper is needed given the information already

available in the literature. Even somewhat disturbing, after carefully reviewing the data presented in Table 1, Table 3, Figures 1-3 and part of Figures 4&5, a majority of the data appears to be repeating what has been presented in the text and SI Fu et al. (2015, ACP) and Wang et al. (2016, ES&T). The discussion provided for these tables and figures are also similar to the arguments provided by the two references. There is little new insight in the discussion of the manuscript.

2. The claim of "serious ecological risks" is an overstatement without clear evidence. The analysis is purely based on potential occurrence of forest fire events and the quantity of Hg storage. In fact, there are few documented cases of Hg pollution of ecological significance caused by forest fires. Should there be fire events, Hg pollution is not likely to be the primary factor leading to negative impacts to the ecosystem. There is no formal risk assessment component in the entire section 5.3 and the discussion in most based on what has been provided in the cited literature.

3. There is little synthesis in the manuscript except Figure 6. Most of the text in the manuscript only re-states the information presented in the figures and tables, rather than providing new insights or specific views of the authors. What is the novelty and what are the new findings in this review?

---

## Author Comment (AC1) · 23 Jul 2018

**Reply to Comments from Reviewer #1**

We thank the editor and reviewers' comments that help us improve the manuscript. We have carefully revised our manuscript following the reviewers' comments. A point-to-point response is given below. The reviewers' comments are in black and our replies and changes in the manuscript are in blue.

**To Reviewer**

***Major Comment:***

The manuscript by Zhou et al. attempts to "provide a better understanding of current knowledge with respect to forest Hg in China and quantify the forest act as net sinks or sources of GEM" and discuss the ecological risk of Hg accumulation in forest ecosystems. Although the authors provide a reasonable summary regarding Hg concentrations in streams with associated "export" fluxes (this is perhaps an ill-defined term by the authors since there is no sufficient evidence that the Hg measured in streams represent "removal" or "export" of Hg from the environmental systems under discussion) and present simplified graphic illustrations for Hg mass balance in various type of forest ecosystems, the manuscript has major deficiencies that do not meet the publication standards of Atmospheric Chemistry and Physics.

Response:

The reviewer think that the current manuscript does not meet the publication standards of *Atmospheric Chemistry and Physics*, but we completely disagreed with the reviewer. This manuscript has provided important information on the current knowledge with respect to forest Hg in China and quantified the forest act as net sinks or sources of GEM, as the Hg emission from the earth surface has a large uncertainty ranges between $-513$ to $1353$ Mg yr$^{-1}$ suggested by a recent review article (Agnan et al., 2016). The review also suggested that the largest uncertainty of natural Hg emission source was resulted from what degree forests are net sinks or sources of GEM. Therefore, the current study focused on the Hg budget in the forest partly help dissolve the question: what degree the ecosystems are net sinks or sources of atmospheric Hg. Additionally, model of Hg dynamics used in North America and Europe could not suit China, because China is the largest emitting country of anthropogenic Hg source and the parameters differed significantly. We find that the Hg retention in forests in China is much higher than the model estimated. Thus we think the manuscript can be potentially published in the *Atmospheric Chemistry and Physics*.

Additionally, the reviewer think that the Hg concentrations in streams with associated export fluxes was incorrected, but all the literatures about Hg export or balance in forest around the world suggested that the stream runoffs was an important export pathway of Hg from forest, such as forested watersheds in Alaska (Vermilyea et al., 2017), forested watershed of the Adirondack Mountains (Gerson et al., 2016), forests in north-central Sweden (Kronberg et al., 2016), all the forests discussed in the manuscript and so on. Therefore, there is no doubt that Hg in streams was an important export fluxes from the forests.

***Comment #1:***

First of all, I am not sure if the review paper is needed given the information already available in the literature. Even somewhat disturbing, after carefully reviewing the data presented in Table 1, Table 3, Figures 1-3 and part of Figures 4&5, a majority of the data appears to be repeating what has been presented in the text and SI Fu et al. (2015, ACP) and Wang et al. (2016, ES&T). The discussion provided for these tables and figures are also similar to the arguments provided by the two references. There is little new insight in the discussion of the manuscript.

Response:

We are completely disagree with the reviewers. The main data in the current manuscript was firstly reviewed (**Table 2&3, Fig. 2 and Fig. 4, 5&6**). Only a little data in the current manuscript overlapped in **Table 1 and Fig. 1&3**, but the data in the current review updates the data in the two studies (Wang et al., 2016; Fu et al., 2015). The data of mercury outputs, soil Hg storage, risks were firstly reviewed in the current study. Details shows in below:

It should be noted that the focus and content in the current manuscript are significantly different from the two researches (Wang et al., 2016; Fu et al., 2015). Wang et al. (2016) assess mercury deposition by litterfall through models, and Fu et al. (2015) mainly discussed the status of atmospheric concentrations based on observations in China and only a simple description on Hg deposition observed in China, which did not focus on forested areas. However, our manuscript was mainly focus on the Hg budgets (input and output) and risks assessment in forested areas of China; therefore, the Hg input by atmospheric depositions must be detailed in the current review.

**In the Table 1 and Fig. 1,** we reviewed the data on the Hg input to the forests of China, we know that some of the data was also reviewed by Fu et al. (2015) and Wang et al. (2016); however, Fu et al. (2015) mainly focused on the higher Hg deposition fluxes associated with higher GEM/TGM concentrations and Wang et al. (2016) focused on the Hg deposition fluxes around the world by models. Additionally, we have updated the data in the two researches. For example, the two researches only reviewed 5 or 6 litterfall deposition fluxes in China, but we have reviewed 22 litterfall deposition fluxes in China. Therefore, we think that the data was not a repetition of previous review and the current review was a more systematic study focused on the Hg input to forested areas in China.

The reviewer also suggested that the data **in Table 3** was also repeating what has been presented in the text and SI Fu et al. (2015) and Wang et al. (2016). However, by carefully reviewing the two researches, we cannot found any data in the Table 3 was reviewed by the two researches. Therefore, the repetition does not exist.

**In Fig. 2**, we found that annual mean atmospheric TGM/GEM concentrations were significantly correlated with the THg concentrations in litterfall samples, and we believed that the data was firstly discussed in the current review. Therefore, no repetition exist.

**Fig. 3** showed correlations between litterfall deposition fluxes of Hg and mass-weighted mean Hg concentrations in litterfall and litterfall biomass. Wang et al. (2015) also showed the correlations, but only **5 pairs of datasets in China** were reviewed in their study and their conclusion was mainly resulted from the data of North America and Europe. The atmospheric Hg concentrations in China was much higher than those in North America and Europe and spatial variation was large in China. Additionally, **the current study reviewed 19 pairs of datasets in China**, which significantly improved the data and conclusion in China.

The reviewer also suggested that part of **Fig. 4 & 5** is repeating what has been presented in the text and SI Fu et al. (2015, ACP) and Wang et al. (2016, ES&T). Through carefully reviewing the data and figures in the two researches, no repetition exist. No similar review of Hg inputs to forest ecosystems in China, Europe and North America (Fig. 4) was found in their studies. In the two reviews (Wang et al., 2016; Fu et al., 2015), there was no content about the soil-atmosphere exchanges.

Therefore, no repetition exist in Fig. 4 & 5.

***Comment #2:***

The claim of "serious ecological risks" is an overstatement without clear evidence. The analysis is purely based on potential occurrence of forest fire events and the quantity of Hg storage. In fact, there are few documented cases of Hg pollution of ecological significance caused by forest fires.

Should there be fire events, Hg pollution is not likely to be the primary factor leading to negative impacts to the ecosystem. There is no formal risk assessment component in the entire section 5.3 and the discussion in most based on what has been provided in the cited literature.

Response:

Firstly, according to the reviewer's suggestion before the manuscript published in the ACPD, the statement of "serious ecological risks" has been revised throughout the manuscript.

Secondly, currently, there was no direct measurement of Hg emission from the wildfires in China and as the reviewer suggested that few documented cases of Hg pollution of ecological significance caused by forest fires, so the detailed about Hg emission from the forest fires was deleted throughout the manuscript.

Thirdly, the reviewer suggested that there was no formal risk assessment component in the entire section, so we have added the risk assessment of Hg intake by consumers. The risk assessment of Hg intake is added as below:

"The chronic dietary intake (CDI, $\mu g\ kg^{-1}\ bw\ day^{-1}$) of Hg depends on both the mushroom Hg concentrations (C) and the daily intake rates (IR), which are widely used to predict the exposure level of humans to trace elements(Du et al., 2016; Zhou et al., 2018). According to the Exposure Factors

Handbook of the US Environmental Protection Agency, the CDI can be calculated as

$$CDI = \sum (C \times IR)/BW \quad (1)$$

where *BW* (kg) is body weight. The IR was assumed as 43 g $day^{-1}$ and the bw was assumed as 60 kg for Chinese residents according to the previous studies in Yunnan province (Kojta et al., 2015;

Falandysz et al., 2015a, b, 2016).

According to the *CDI* of mushroom consumptions, a Hazard Quotient (*HQ*) indicating the noncarcinogenic health risk during a lifetime can be calculated by dividing the *CDI* by the toxicity threshold value of the reference dose (*RfD*).

$$HQ = CDI/RfD \quad (2)$$

The recommended *RfD* of Hg by Joint Food and Agriculture Organization (FAO)/WHO Expert Committee on Food Additives is 0.57 µg kg$^{-1}$ bw day$^{-1}$ (JECFA 2010). When the *HQ* is ≤ 1, the adverse health effects are unlikely experienced, whereas the value > 1 indicates potential non-carcinogenic effects. Based on the averaged Hg concentrations in the mushrooms from five studies in subtropical forests of China, all the values of HQ showed > 1, demonstrating a much higher non-carcinogenic risk to consumers caused by daily intake of Hg through mushroom ingestions (Table S2). The result suggested the need for greater focus on the adverse health effects induced by Hg on the consumers in China."

**See the revised manuscript, line 644-662 and Table S2 in Supporting Information.**

**Table S2.** Hg concentrations (µg g$^{-1}$) in mushrooms and dietary intake risks in China.

| Site | Species | Altitude (m a.s.l) | Location Type | Study period | THg concentration | IR | CDI | HQ | References |
|---|---|---|---|---|---|---|---|---|---|
| Mt. Gongga | *Gymnopus erythropus* | 2947.8 | Remote | September, 2012 | 2.36 | 43 | 1.67 | 2.9 | Falandysz et al., 2014 |
| Mt. Gongga | *Marasmius dryophilus* | 2947.8 | Remote | September, 2012 | 0.87 | 43 | 0.62 | 1.1 | Falandysz et al., 2014 |
| Yunnan Province | Fungi genus *Xerocomus* | 2000-4200 | Remote | 2011–2014 | 0.86 | 43 | 0.62 | 1.1 | Kojta et al., 2015 |
| Mt. Gongga and Yunnan | 27 species | 2000-4200 | Remote | 2012–2014 | 1.48 | 43 | 1.06 | 1.86 | Falandysz et al., 2016 |
| Yunnan Province | Genus *Leccinum* | | Remote | 2011–2014 | 2.13 | 43 | 1.53 | 2.67 | Falandysz et al., 2015b |
| Yunnan Province | Macrocybe gigantea | | Remote | 2011-2013 | 1.05 | 43 | 0.75 | 1.3 | Wiejak et al., 2014 |

At last, the discussion based on what has been provided in the cited literature was deleted and the data was summarized in the section 5. See the revised manuscript, line 625-668.

*Comment #3:*

There is little synthesis in the manuscript except Figure 6. Most of the text in the manuscript only re-states the information presented in the figures and tables, rather than providing new insights or specific views of the authors. What is the novelty and what are the new findings in this review?

Response:

We completely disagree with the reviewer. Only a little text in the manuscript re-states the information presented in the figures and tables and the main text are discussing the status of Hg in forested areas of China.

**In the section 2.** Processes of Hg input, only about 600 words are showing the results of Hg input to forests, while the total number of words is 2216. Additionally, the text of the 600 words were mainly in statistical analysis of the data in the figures and tables, but not re-state the information presented in the figures and tables as the reviewer suggested. The other 1600 words in this section are discussing why higher Hg input in China and the difference with North America and Europe.

**In the section 3**. Processes of Hg output, only about 300 words shows the results of Hg output from the forests, while the total number of words is 2286 in this section. Therefore, we believe that the main text does not re-state the information presented in the figures and tables, while the main text of this section is discussing the machine of Hg export from the forests.

**In the section 4**. Hg budgets, the main text shows the synthesis of section 2 and 3, so not any restatements of the information presented in the figures and tables is in this section.

**In the section 5.** Hg storage and risk assessment, about 400 words are showing the results of Hg storage in the forests, while the total number of words is 1716. Same as we stated above, the text of the 400 words were mainly in statistical analysis of the data in the figures and tables. In this section, we have compared the Hg pools in China with those in North America and Europe and presented the risks of Hg storages in the forest by a model.

**In the section 6.** Environmental implication and research needs **and 7.** Conclusions**, both of the sections are summary, no information in these sections presented in the figures and tables, so no re-statements exist.

**The novelty of the current review** is that it has answered the question of what degree the forest ecosystems are net sinks or sources of atmospheric Hg, which was raised by a recent review article (Agnan et al., 2016). Agnan et al. (2016) showed that the Hg emission from the earth surface has a large uncertainty ranges between −513 to 1353 Mg yr$^{-1}$ and the uncertainty was mainly from the forest, because forest acted as net sinks or sources of atmospheric Hg is unresolved.

**The new findings in this review** show below: **Firstly**, model of Hg dynamics used in North America and Europe dose not suit for China, because China is the largest emitting country of anthropogenic Hg source and the parameters differed significantly. A large underestimation of model estimation compared to the observation-based estimation from forest areas of China, and the current review suggests future model studies should consider the THg dry deposition in forested areas individually. **Secondly**, previous study showed forest ecosystems appear a net deposition of 59 t yr$^{-1}$. However, based on the field observations of THg retention in Chinese forests, the current review roughly estimates the THg retention in forest soils was 69 t yr$^{-1}$ just in China, which was much higher than the global data of 59 t yr$^{-1}$. **Thirdly,** the large uncertainties of estimations by models are mainly resulted from the variation of reported atmospheric Hg uptake by foliage and the limited geospatial representation of available data, more studies on the Hg budget in forest are needed. **Fourthly**, the large ''active'' soil pool at forests is a potential short-term and long-term source of THg and MeHg to downstream aquatic ecosystems; however, there is no study reporting the accumulation of THg and MeHg in aquatic ecosystem after output from the forest ecosystem in China, the studies of which are needed.

**The detailed discussion shows in 670-703 in the manuscript.**

---

## Author Comment (AC2) · 23 Jul 2018

**Reply to Comments from Reviewer #2**

We thank the editor and reviewers' comments which help us to improve the manuscript. We have carefully revised our manuscript following the reviewers' comments. A point-to-point response is given below. The reviewers' comments are in black and our replies are in blue.

**To reviewer**

*Comment #1:*

This manuscript reviewed the mercury fluxes, budgets and pools in forest ecosystems in China, however, the authors did not do a good systemic summary for all the current research. For example, the underground flow was not considered into the budget.

Response:

We have revised the manuscript by around and around and revised all the comments. First, the fluxes from the underground flow were estimated in the manuscript and was considered into the budget, showing Section 3.1, Section 4 and Fig. 6.

The underground flow considered into the budget is revised as below:

"The direct measurements of THg in underground runoffs were not conducted in any forests of China, but they played important roles in the THg export from forests due to both of the amounts and THg concentrations usually higher than those of surface runoffs in subtropical forests (Liu, 2005; Luo et al., 2015b). Several studies have measured THg concentrations in solutions of soil profiles in subtropical forest of Tieshanping, which was averaged 21.8 ng $L^{-1}$ and ranged from 1.98 to 60 ng $L^{-1}$ (Wang et al., 2009; Zhou et al., 2015; Luo et al., 2015b). The observed THg concentrations of soil solution was higher than those in five Swiss forest soils, and the reason may be due to higher THg loads and soil THg content in this Chinses forest. Although no studies directly measured the export flux of THg via underground runoff, we roughly estimated the flux based on the THg in soil solutions and runoff amount in Tieshanping forest, which is 6.0 µg $m^{-2}$ $yr^{-1}$; therefore, the total Hg output by runoffs as the sum of Hg output by surface runoff (3.5 µg $m^{-2}$ $yr^{-1}$) and underground runoff (6.0 µg $m^{-2}$ $yr^{-1}$) was 9.5 µg $m^{-2}$ $yr^{-1}$." and "Based on our measured THg concentrations in soil solution (9.2 ng $L^{-1}$, our unpublished data) and the amounts of underground runoffs in the three stands (Wang et al., 2012), the export fluxes by underground runoffs were estimated."

[Figure]

[Figure]

**Fig. 6.** Total mercury budgets (µg m$^{-2}$ yr$^{-1}$) at the four subtropical forests of Tieshanping (I), Qianyanzhou (II), Mt. Gongga (III) and Mt. Simian forests (IV) (a) and three temperate forest stands of Mt. Dongling (I-III) (b). LF, TF, SR and UR represent litterfall, throughfall, surface runoff and underground runoff fluxes, respectively. Data are from Zhou et al. (2016a, c), Luo et al. (2015b), Wang et al. (2009), Luo et al. (2015a), Fu et al. (2010a), Ma et al., 2016.

*Comment #2:*

There are numerous typographical errors and poor sentence structure throughout the paper.

Response:

We have revised the typographical errors and poor sentence structure throughout the paper by us and our colleagues. We hope the revised grammar and sentence structure meet the publish standard.

***Comment #3:***

Tieshanping showed a high Hg pool, why? , due to its location or being influenced by human activities.

Response:

The high Hg pool in Tieshanping is due to it located near the larger city of Chongqing, which has emitted large amount of atmospheric Hg annually and showed as "The much higher Hg input at Tieshanping forest is due to it located near the center of Chongqing City (20 km), the annual atmospheric emissions of which just from coal combustion was 4.97 t (Wang et al., 2006) and Hg pollution was regarded as major environmental burdens in Chongqing (Yang et al., 2009). The large mercury emission resulted in much higher Hg deposition fluxes not only in the urban areas but also in the suburban areas (Ma, 2015; Wang et al., 2009, 2014)." in line 311-315.

***Comment #4:***

Line 29, Chinses??

Response:

It has changed to "Chinese" in line 179.

***Comment #5:***

Line 43, change are to is

Response:

It has changed to "is" accordingly in line 195.

***Comment #6:***

Line 50, change sinks to sink

Response:

It has changed accordingly in line 202.

***Comment #7:***

Line 120, change plays to play

Response:

It has changed accordingly in line 278.

***Comment #8:***

Line 136, change resulted to result; which is closed to the large Hg mine of Wanshan, I think your mean is that the litterfall Hg concentration at Leigong is closed to the Hg concentration of litterfall in

Wanshan?? If like this, please reorganize this sentence.

Response:

We have changed the sentence as "Although Mt. Leigong was relatively isolated from anthropogenic activities with lower GOM, PBM, precipitation and throughfall Hg concentrations,

GEM could undergo long-range transport from emission sources and the GEM concentration was 2.80

ng m−3 in Mt. Leigong that is about 170 km to the large Hg mine of Wanshan (Fu et al., 2010a). The relatively higher GEM concentration resulted in elevated litterfall Hg concentrations." in line 291-294.

***Comment #9:***

Line 152, change we to it

Response:

It has changed accordingly in line 307.

***Comment #10:***

Line 154, change was to were

Response:

It has changed accordingly in line 310.

***Comment #11:***

Line 157, change ranging to range

Response:

It has changed accordingly in line 317.

***Comment #12:***

Line 209, change has to have

Response:

It has changed accordingly in line 366.

*Comment #13:*

Line 214, change depends to depend

Response:

It has changed accordingly in line 371.

*Comment #14:*

Line 223, only one sites??

Response:

It has changed to "only one site" in line 390.

*Comment #15:*

Line 239, change were to was

Response:

It has changed accordingly in line 404.

*Comment #16:*

Line 247, change showed to show

Response:

It has changed accordingly in line 412.

*Comment #17:*

Line 277, humidity??

Response:

It has changed to "humidity" in line 442.

*Comment #18:*

Line 287, change were to was

Response:

It has changed accordingly in line 451.

*Comment #19:*

Line 306, add were after forests

Response:

It has added in line 470.

*Comment #20:*

Line 439, change was to were

Response:

It has revised accordingly.

**Mercury fluxes, budgets and pools in forest ecosystems of China: A critical review**

Jun Zhou [a, b, e,] *, Buyun Du [c], Zhangwei Wang [d], Lihai Shang [c], Xingjun Fan [b], Jing Zhou [a, e,] *

a. Key Laboratory of Soil Environment and Pollution Remediation, Institute of Soil Science, Chinese Academy of Sciences, Nanjing 210008, China.

b. College of Resource and Environment, Anhui Science and Technology University, Fengyang, Anhui 233100, P. R. China c. 
[revised manuscript text omitted]
, 2016). The chronic dietary intake (CDI, µg kg$^{-1}$ bw day$^{-1}$) of Hg depends on both the mushroom Hg concentrations (C) and the daily intake rates (IR), which are widely used to predict the exposure level of humans to trace elements(Du et al., 2016; Zhou et al., 2018). According to the Exposure Factors Handbook of the US Environmental Protection Agency, the CDI can be calculated as

$$CDI = \sum (C \times IR)/BW \quad (1)$$

where $BW$ (kg) is body weight. The IR was assumed as 43 g day$^{-1}$ and the bw was assumed as 60 kg for Chinese residents according to the previous studies in Yunnan province (Kojta et al., 2015).

According to the *CDI* of mushroom consumptions, a Hazard Quotient (*HQ*) indicating the non-carcinogenic health risk during a lifetime can be calculated by dividing the *CDI* by the toxicity threshold value of the reference dose (*RfD*).

$$HQ = CDI/RfD \quad (2)$$

The recommended *RfD* of Hg by Joint Food and Agriculture Organization (FAO)/WHO Expert Committee on Food Additives is 0.57 µg kg$^{-1}$ bw day$^{-1}$ (JECFA 2010). When the *HQ* is ≤ 1, the adverse health effects are unlikely experienced, whereas the value > 1 indicates potential non-carcinogenic effects. Based on the averaged Hg concentrations in the mushrooms from five studies in subtropical forests of China, all the values of HQ showed > 1, demonstrating a much higher non-carcinogenic risk to consumers caused by daily intake of Hg through mushroom ingestions (Table S2). The result suggested the need for greater focus on the adverse health effects induced by Hg on

[revised manuscript text omitted]

Iverfeldt, Å.: Mercury in forest canopy throughfall water and its relation to atmospheric deposition, Water Air Soil Poll., 56, 553–564, 1991.

JECFA, (Joint FAO/WHO Expert Committee on Food Additives). (2010). Joint FAO/WHO food standards programme, committee of the codex alimentarius commission, 33rd session. Geneva, Switzerland, July 5–9.

[revised manuscript text omitted]

Wang, Y.: Study on eco-hydrological process to Land use/forest cover change of small typical watersheds in Beijing mountain area, Doctor's dissertation, Beijing Forestry University, 2012 (in Chinese with English abstract).

Wang, Z. W., Zhang, X. S., Xiao, J. S., Zhijia, C., and Yu, P. Z.: Mercury fluxes and pools in three subtropical forested catchments, southwest China, Environ. Pollut., 157, 801–808, doi:10.1016/j.envpol.2008.11.018, 2009.

Wright, L. P., Zhang, L., and Marsik, F. J.: Overview of mercury dry deposition, litterfall, and throughfall studies, Atmos. Chem. Phys.,
16(21), 1–46, 2016.

Xiao, Z. F., Munthe, J., W. H. S., and Lindqvist, O.: Vertical fluxes of volatile mercury over forest soil and lake surfaces in Sweden,
Tellus B, 43(3), 267–279, 1991.

Xiao, Z., Sommar, J., Lindqvist, O., and Giouleka, E.: Atmospheric mercury deposition to grass in southern Sweden, Sci. Total Environ.,
213(213), 85-94, 1998.

Xin, M. and Gustin, M. S.: Gaseous elemental mercury exchange with low mercury containing soils: Investigation of controlling factors,
Appl. Geochem., 22, 1451–1466, 2007.

Xue, T., Wang, R. Q., Zhang, M. M., and Dai, J. L.: Adsorption and desorption of mercury (II) in three forest soils in Shandong province,
China, Pedosphere, 23(2), 265–272, 2013.

Yang, Y. K., Chen, H., and Wang, D. Y.: Spatial and temporal distribution of gaseous elemental mercury in Chongqing, China. Environ.
Monit. Assess. 156, 479-489, 2009.

Yin, Y., And, H. E. A., Huang, C. P., Sparks, D. L., and Sanders, P. F.: Kinetics of mercury (ii) adsorption and desorption on soil, Environ.
Sci. Technol., 31(2), 496–503, 1997.

Zhang, H. and Lindberg, S. E.: Sunlight and iron (III)-induced photochemical production of dissolved gaseous mercury in freshwater,
Environ. Sci. Technol., 35, 928–935, 2001.

Zhang, H., Fu, X., Lin, C. J., Shang, L., Zhang, Y., Feng, X., and Lin, C.: Monsoon-facilitated characteristics and transport of atmospheric
mercury at a high-altitude background site in southwestern China, Atmos. Chem. Phys., 16(20), 1-36, 2016.

Zhang, H., Lindberg, S. E., Barnett, M. O., Vette, A. F., and Gustin, M. S.: Dynamic flux chamber measurement of gaseous mercury
emission fluxes over soils. Part 1: simulation of gaseous mercury emissions from soils using a two-resistance exchange interface
model, Atmos. Environ., 36, 835–846, 2002.

Zhang, H., Lindberg, S. E., Marsik, F. J., and Keeler, G. J.: Mercury air/surface exchange kinetics of background soils of the
Tahquamenon River watershed in the Michigan Upper Peninsula, Water, Air, Soil Pollut., 126 (1−2), 151−169, 2001.

Zhou, J., Feng, X., Liu, H., Zhang, H., Fu, X., Bao, Z., Wang, X., and Zhang, Y.: Examination of total mercury inputs by precipitation
and litterfall in a remote upland forest of southwestern China, Atmos. Environ., 81, 364–372, doi:10.1016/j.atmosenv.2013.09.010,
2013a.

Zhou, J., Lang, X., Du, B., Zhang, H., Liu, H., Zhang, Y., and Shang L.: Litterfall and nutrient return in moist evergreen broad-leaved
primary forest and mixed subtropical secondary deciduous broad-leaved forest in China, Eur. J. Forest Res., 135(1), 77–86, 2016b.

Zhou, J., Liang, J., Hu, Y., Zhang, W., Liu, H., You, L., Zhang, W., Gao, M., Zhou, J.: Exposure risk of local residents to copper near the
largest flash copper smelter in china. Sci. Total Environ., 630, 453-461, 2018.

[revised manuscript text omitted]